# Comparison of serum lactate and lactate-derived ratios as prognostic biomarkers in pediatric dengue shock syndrome using supervised machine learning models

Nguyen Tat Thanh[1,2]*, Vo Thanh Luan[1]

**1** Department of Infectious Diseases, Children Hospital 2, Ho Chi Minh City, Vietnam, **2** TB Department, Woolcock Institute of Medical Research, Ho Chi Minh City, Vietnam

* thanhhonor@gmail.com

## Abstract

### Background

Dengue shock syndrome (DSS), with critical complications encompassing mechanical ventilation (MV), dengue-associated acute liver failure (PALF), and encephalitis, is associated with high mortality in children. Although serum lactate is a recognized prognostic biomarker, it may not fully reflect the complex metabolic disturbances in DSS. Recent evidence suggests that lactate-derived indices, including lactate-to-albumin ratio (LAR) and lactate-to-bicarbonate ratio (LB), may enhance prognostic accuracy. This study aimed to evaluate and compare the predictive performance of the LAR, LB ratio, and serum lactate levels in pediatric DSS using machine learning approaches.

### Methods and findings

We conducted a secondary analysis of a retrospective cohort study involving children with DSS at a tertiary pediatric center in Vietnam (2013–2022). The primary composite endpoint included in-hospital mortality, MV, dengue-associated PALF and encephalitis. Predictors were selected based on clinical expertise, literature review, Akaike Information Criterion and Least Absolute Shrinkage and Selection Operator. Multiple supervised machine-learning algorithms – logistic regression, random forest (RF), support vector machine (SVM), k-nearest neighbor, naïve Bayes, AdaBoost, and XGBoost - were applied. Model performance was evaluated using the area under the receiver operating characteristic curve (AUC) and feature importance was assessed using Shapley Additive Explanations (SHAP).

**Data availability statement:** Data availability: All relevant data are within the manuscript and its Supporting Information files. Our primary data source: All relevant data supporting the findings of this study are available within the paper, its Supporting Information files, and the primary dataset from the parent study: Nguyen Tat T, Vo Hoang-Thien N, Nguyen Tat D, Nguyen PH, Ho LT, Doan DH, et al. Prognostic values of serum lactate-to-bicarbonate ratio and lactate for predicting 28-day in-hospital mortality in children with dengue shock syndrome. Medicine (Baltimore). 2024;103(17):e38000. doi: 10.1097/MD.0000000000038000. PMID: 38669370; PMCID: PMC11049702. This dataset was approved by the Scientific Committee and Institutional Review Board of Children's Hospital No.2, Ho Chi Minh City, Vietnam (IRB No. 893/QD-BVND2; 06 June 2022).

**Funding:** The author(s) received no specific funding for this work.

**Competing interests:** The authors have declared that no competing interests exist.

**Abbreviations:** DSS, Dengue shock syndrome; MV, Mechanical ventilation; PALF, Dengue-associated pediatric acute liver failure; PICU, Pediatric intensive care unit; ML, Machine learning; LASSO, Supervised models, the Least Absolute Shrinkage and Selection Operator; LR, Logistic regression; RF, Random forest; KNN, k-nearest neighbors; SVM, Support vector machine; XGBoost, eXtreme Gradient Boost; SHAP, Shapley Additive Explanations.

## Results

Of the 524 eligible patients (median age: 8.7 years), 17% met the composite end-point. At admission, LAR demonstrated superior discriminatory ability (AUC: 0.82; 95% CI: 0.76–0.87) compared to serum lactate (AUC: 0.72; 95% CI: 0.65–0.78) and LB ratio (AUC: 0.68; 95% CI: 0.62–0.74) (all $p < 0.001$). The Youden-index based optimal LAR cutoff was 1.25, whereas that for the LB ratio was 0.20. The RF, XGBoost and SVM models achieved the highest performance. SHAP analysis revealed that LAR was the most influential predictor among the lactate-based variables.

## Conclusions

LAR exceeded serum lactate and the LB ratio in predicting critical outcomes in pediatric DSS. These findings support its utility as a practical and accessible tool for early risk stratification in DSS patients. These results support the use of LAR as a practical and accessible tool for risk stratification in pediatric dengue care.

## Introduction

Dengue hemorrhagic fever remains a significant public health challenge in tropical and subtropical regions, with a disproportionately high burden on children in Southeast Asia and Latin America. According to a 2024 World Health Organization (WHO) report, the global dengue incidence has surged to unprecedented levels, with over 7.6 million cases, 16,000 severe cases, and 3,000 deaths reported [1]. Among hospitalized pediatric populations, mortality rates range from 5% to 25%, predominantly because of severe disease complications, such as hypovolemic shock, severe bleeding, progressive plasma leakage, acute liver injury, and respiratory failure [2,3]. Given the absence of disease-specific antiviral therapies, early identification of high-risk patients is essential for timely intervention and reduction in morbidity and mortality. However, data on prognostic markers in critically ill children with dengue shock syndrome (DSS) are insufficient.

Lactic acidosis is a frequently observed metabolic disturbance in DSS, reflecting impaired tissue perfusion, hypoxia, and hepatic dysfunction [4]. Dengue-induced hepatic dysfunction can contribute to increased serum lactate and reduced bicarbonate levels, both of which are associated with poor clinical outcomes [4]. Hypoalbuminemia is a common feature in severe dengue, contributing to vascular leakage, circulatory instability, and increased mortality risk [5,6]. Hence, elevated lactate levels reflect anaerobic metabolism and tissue hypoperfusion, while low serum albumin levels may indicate systemic inflammation, fluid redistribution, and liver dysfunction, which are central to the pathophysiology of dengue shock. Notably, no single biomarker encompassing serum lactate, albumin, and bicarbonate may sufficiently reflect the complex pathophysiology of DSS, particularly when influenced by massive hepatocyte damage, endothelial injury, heavy coagulation disorders, and acid-base disturbances [7]. Recent studies have proposed combining lactate with other parameters, such as albumin or bicarbonate, to form ratios, specifically the lactate-to-albumin

ratio (LAR) and lactate-to-bicarbonate (LB) ratio, which may improve prognostic utility [4]. The lactate-to-albumin ratio integrates these two markers and has shown prognostic value in critical diseases, such as sepsis [8,9]. Similarly, the lactate-to-bicarbonate ratio has emerged as a surrogate for the severity of metabolic acidosis and has been associated with an increased mortality risk in dengue-related shock [4].

However, there is a paucity of pediatric data evaluating the prognostic utility of the LAR and LB ratio in predicting clinical outcomes and in-hospital mortality among children diagnosed with dengue shock syndrome [4]. Furthermore, the comparative performance of these lactate-derived ratios in pediatric DSS remains understudied, particularly in low- and middle-income settings. To date, no existing studies have employed advanced analytical techniques, such as supervised machine learning, to evaluate the prognostic utility of these biomarkers within a clinically meaningful framework in patients with severe dengue and DSS. This study aimed to assess and compare the prognostic performance of serum lactate, LAR, and LB ratio in predicting critical in-hospital outcomes in children with DSS. Machine learning models have also been employed to study the contributions of these biomarkers within a broader clinical context.

## Methods

### Data source, study setting and population

This study employed a secondary data analysis obtained from a previously approved investigation entitled *"Prognostic model predicting mortality among patients presenting with dengue shock syndrome at Children's Hospital 2, during 2013–2022"* [4]. Ethical approval for the parent study was granted by the Scientific Committee and Institutional Review Board (IRB) of Children's Hospital 2, Ho Chi Minh City, Vietnam (IRB approval number. 893/QD-BVND2; 06/06/2022) [4]. The requirement for informed consent was waived, and participants' identities were anonymized to ensure patient confidentiality. The study was performed in accordance with the principles of Good Clinical Practice and ethical guidelines of the Declaration of Helsinki. The original dataset was accessed and analyzed on 01/05/2025. The current report adheres to the Transparent Reporting of a Multivariable Prediction Model for Individual Prognosis or Diagnosis (TRIPOD) guidelines (S1 Checklist).

This was a retrospective, single-center study conducted at Children's Hospital 2, with the capacity of a 1,400-bed tertiary pediatric referral center serving pediatric patients across the central and southern regions of Vietnam. This study included pediatric patients diagnosed with dengue shock syndrome who were admitted between 2013 and 2022. The primary objective was to assess and compare the prognostic performance of serum lactate, the lactate-to-albumin ratio and lactate-to-bicarbonate ratio in predicting a composite clinical endpoint comprising in-hospital mortality, the need for mechanical ventilation, and dengue-associated acute liver failure and encephalitis. Eligible participants were children under 18 years of age with a clinical diagnosis of dengue shock syndrome and laboratory-confirmed dengue infection [10]. Patients were excluded if dengue infection was not serologically confirmed or if more than 20% of clinical or laboratory data were missing.

### Study definitions

In accordance with the 2009 WHO dengue guidelines [10], laboratory-confirmed dengue infection was defined as the presence of dengue-specific IgM antibodies or a positive nonstructural protein 1 (NS1) antigen test. Diagnoses of dengue shock syndrome and severe hemorrhagic manifestations were establish based on the WHO 2009 dengue guidelines [10]. Severe bleeding was defined as hemorrhagic events leading to hemodynamic compromise, necessitating urgent hemostatic interventions and blood product transfusions. Clinical manifestations included hemoptysis, hematemesis, melena, hematochezia, and hematuria. Severe transaminitis was defined as elevated aspartate aminotransferase (AST) and/or alanine aminotransferase (ALT) levels ≥ 1,000 IU/L [10].

**Dengue-associated acute liver failure** (PALF) was diagnosed based on the criteria established by the European Association for the Study of the Liver [11]. It is characterized by acute severe hepatic dysfunction in children with no prior

history of chronic liver disease. Diagnostic criteria included evidence of significant hepatocellular injury with coagulopathy unresponsive to vitamin K, defined as an international normalized ratio (INR) > 1.5 in the presence of hepatic encephalopathy or INR > 2.0 in its absence.

**Dengue-associated encephalitis** was diagnosed in patients with a sustained alteration in mental status lasting over 24 h, manifested by confusion, diminished consciousness, coma, or behavioral abnormalities, alongside the detection of dengue virus-specific markers (RNA, NS1 antigen, or IgM antibodies) in cerebrospinal fluid (CSF) and associated CSF monocytosis, in the absence of alternative etiologies [12,13].

**Indications for mechanical ventilation (MV) in children with severe dengue.** In dengue-infected patients presenting with acute respiratory distress, our initial intervention involved the application of nasal continuous positive airway pressure. If this intervention proves insufficient, invasive mechanical ventilation is subsequently initiated [2]. Further indications for mechanical ventilation include severe dengue complicated by large-volume pleural or peritoneal effusions, abdominal compartment syndrome, pulmonary edema, fluid overload, acute respiratory distress syndrome, or the requirement for high-rate intravenous fluid administration (≥ 7 mL/kg/h) sustained over the first 12 h of pediatric intensive care unit (PICU) admission. Furthermore, MV is used in cases of dengue-associated encephalitis when patients are unstable on supplemental oxygen or present with episodes of apnea [2].

## Study outcome

The primary outcome of the study was the composite clinical endpoint, which included in-hospital death, mechanical ventilation requirement, dengue-associated acute liver failure and encephalitis.

## Candidate variables

From the original dataset, 524 eligible participants with 75 variables, were identified [4]. All variables were assessed for completeness, clinical relevance, and consistency with the known disease pathophysiology. A total of 23 variables were selected as candidate predictors (S1 Table). These variables were selected based on their biological plausibility and relevance to dengue pathogenesis. The selected variables were categorized as follows.

- **Demographic variables**: age, sex, and presence of underlying comorbidities.

- **Clinical variables**: severity and timing of DSS onset, severe bleeding, platelet transfusion requirement, neurological status (AVPU scale), respiratory rate, systolic and diastolic shock indices, vasoactive inotropic score (VIS), and cumulative fluid volume administered from referring hospital and within the first 24 h of PICU admission.

- **Laboratory variables**: hematological indices, liver transaminases, serum lactate, serum albumin, serum bicarbonate, and derived ratios, including the LAR and LB ratio.

## Data measurement

The LAR and LB ratio were calculated using the following formulae:

- LAR = serum lactate (mmol/L)/ serum albumin (g/dL)

- LB ratio = serum lactate (mmol/L)/ serum bicarbonate (mEq/L)

## Feature selection and data preprocessing

Feature selection was guided by both clinical expertise and statistical modelling. A hybrid approach combining the Akaike Information Criterion (AIC) and Least Absolute Shrinkage and Selection Operator (LASSO) regression was used to reduce dimensionality and mitigate multicollinearity. The selected variables were included in subsequent model training.

Data preprocessing included the conversion of categorical variables into factors and binary clinical outcomes (e.g., death, mechanical ventilation, pediatric acute liver failure, and encephalitis) into a numeric (0/1) format. Implausible values and irrelevant categories were removed as part of the data-cleaning process.

### Handling of missing data

The extent and pattern of missing data were presented with frequencies and proportions (%) and displayed in a Pareto chart (S1 Fig). Variables with more than 20% missingness were excluded. For the remaining variables, multiple imputations were performed using the predictive mean matching with the *missRanger* package in R (version 4.5.0) [14]. The imputed dataset was then used for all machine learning analyses.

### Conventional statistical analysis

Continuous variables were summarized using medians and interquartile ranges (IQRs), while categorical variables were presented as counts and percentages. Distributions of the key clinical and biomarker variables are illustrated in S2 Fig. Standardized mean differences (SMD) were computed for all covariables using the *tableone* package in R (version 4.5.0) to evaluate distributional balance between outcome groups (S2 Table). ROC curves were generated to evaluate individual biomarkers, and area under the curve (AUC) metrics were calculated for lactate, LAR, and LB ratio. These metrics were also used to assess the performance of the multivariable predictive models.

### Machine learning model development

All the machine learning procedures are summarized in a study analysis flowchart (S1 File).

**Model selection.** Various supervised machine learning algorithms were implemented: logistic regression (LR), random forest (RF), support vector machine (SVM), k-nearest neighbors (KNN), AdaBoost, eXtreme Gradient Boosting (XGBoost), and Naïve Bayes. These models were chosen for their widespread application and robust performance in clinical classification tasks.

**Data partitioning and transformation.** The dataset was randomly partitioned into training (80%) and testing (20%) sets using stratified sampling to maintain outcome balance. Categorical features were one-hot encoded and continuous variables were normalized using z-score standardization.

**Model training, hyperparameter tuning and validation.** Model training was conducted using 5-fold cross-validation of the training set to minimize overfitting. Hyperparameter tuning was performed for supervised models using a grid search strategy with 5-fold cross-validation to minimize overfitting and optimize generalizability. For the random forest, the number of trees, maximum tree depth, and minimum number of samples required for node splitting were tuned. For the support vector machine, the penalty parameter (C), kernel type, and kernel-specific parameters (gamma for radial basis function kernels) were adjusted. For the XGBoost model, the tuning included the learning rate, number of boosting iterations, maximum tree depth, and subsampling ratios. The optimal hyperparameters were selected based on cross-validated performance metrics. The performance of the final model was assessed on an independent test set using standard classification metrics.

**Model performance evaluation.** Model performance was assessed using the accuracy, sensitivity, specificity, precision, F1 score, and AUC. These metrics provided a comprehensive evaluation of model discrimination and calibration across all classifiers.

### Model interpretation and feature importance

To enhance the model interpretability, the Shapley Additive Explanations (SHAP) framework was applied to the optimal predictive models. The SHAP values quantify the marginal contribution of each predictor to the model output, offering transparent insight into the feature importance and directionality of the effect.

## Decision curve analysis

The decision curve analysis was performed to analyze the net clinical benefit across a broad range of threshold probabilities for predictive models.

## Software and computational environment

All statistical analyses were conducted using R software (version 4.5.0) and Python (Anaconda Distribution version 3.0). In R, baseline characteristics and group comparisons were summarized with descriptive and tabulation packages (table 1, *compareGroups*, *tableone*, *flextable*, *gtsummary*, *tidyverse*, *dplyr*, *epiDisplay*). Predictive values and cutoff points were assessed with *pROC*, *cutpointr*, *epiR*, and *binom*. Missing data were addressed through imputation methods (*naniar*, *VIM*, *missRanger*). Supervised predictive modeling employed regression- and tree-based approaches using *glmnet*, *leaps*, *MASS*, *pls*, *caret*, *caTools*, *randomForest*, *mlr*, *e1071*, and *xgboost*. Model performance was evaluated using ROC analysis, calibration (*Hmisc*, *rms*), and decision-curve analysis (*rmda*), while interpretability was assessed with variable importance measures and SHAP values (*vip*, *SHAPforxgboost*, *iml*). In Python, additional models were trained using *scikit-learn*, with ensemble boosting performed via *xgboost* and *lightgbm*. Model explainability was supported by the *shap* package, and data preprocessing and visualization were conducted using *numpy*, *pandas*, *matplotlib*, and *seaborn*.

## Results

### Baseline characteristics of study participants upon PICU admission

Between 2013 and 2022, approximately 2,000 pediatric patients with DSS were admitted to the PICU, of whom 524 eligible patients who had available clinical and biomarker data retrieved were included in the analysis (Fig 1). Clinical and laboratory characteristics of the study cohort at the time of admission are presented in Table 1. The median age of the patients was 8.7 years (interquartile range [IQR]: 6–11 years), with females comprising approximately 44% of the cohort. The median body mass index was 18.9 kg/m² (IQR: 16.3–22.5). A majority of patients (88%) presented with compensated dengue shock syndrome (DSS), while 12% were diagnosed with decompensated DSS. Severe bleeding was observed in 55 (10.5%) patients. Upon PICU admission, the median systolic shock index was 1.3 bpm/mmHg (IQR: 1.1–1.5). Patients with critical dengue-related complications required higher vasoactive inotropic support, with the median of VIS within first 6h of admission was 15 (IQR: 0–30). Patients who experienced the composite endpoint exhibit more impaired neurological status as assessed by the AVPU scale and presented with higher respiratory rate. Hematological analysis revealed a marked elevation in hematocrit levels accompanied by a reduction in platelet count. Approximately one-third of patients had low platelet counts < 20 x10⁹/L. Severe transaminitis was observed in 74 (14.1%) patients. An elevated international normalized ratio (INR) was noted, with a median value of 1.28 (IQR: 1.15–1.62). The median serum creatinine concentration was 53 μmol/L (IQR: 45–61 μmol/L). At admission, the median serum lactate was 2.3 mmol/L (IQR:1.7–3.2), whereas that of serum bicarbonate was 17.1 (IQR: 14.7–19.4) mEq/L. The median serum albumin was 2.8 (IQR: 1.9–3.4) g/dL. Hence, the median LAR was 0.86 (IQR: 0.61–1.3), and that for LB ratio was 0.14 (IQR: 0.09–0.23). Notably, there were signficant differences in the baseline variables among patients with and without composite endpoint, including age, early onset of shock, DSS severity, severe bleeding, shock indices, neurological status, respiratory rate, laboratory values, LAR and LB. Baseline characteristics stratified by composite endpoint, with standardized mean differences to assess distributional balance between groups are detailed in S2 Table.

In this cohort, 22 patients with dengue shock syndrome (4.2%) died during their stay in the PICU and all deaths belonged to the patients with composite endpoint. The main causes of mortality included dengue-associated acute liver failure, acute respiratory distress syndrome, pulmonary and cerebral hemorrhage, and multi-organ failure. Notably, 85 patients (16.2%) experienced severe respiratory failure requiring mechanical ventilation support, 41 (7.8%) were diagnosed with dengue-associated acute liver failure, and 4 (0.8%) were diagnosed with dengue-associated encephalitis.

**Table 1. Baseline clinical and laboratory characteristics of study participants on admission and clinical outcomes at discharge.**

| Characteristics | All participants (N = 524) | Composite endpoint (+) (n = 89) | Composite endpoint (-) (n = 435) | p-value |
|---|---|---|---|---|
| Age (years) | 8.7 (6-11) | 7 (4.2-9.1) | 9 (6.5-11.4) | < 0.001 |
| Female patients, n (%) | 232 (44) | 40 (45) | 192 (44) | 0.98 |
| Body mass index (kg/m$^2$) | 18.9 (16.3-22.5) | 18.9 (16.5-22.5) | 18.9 (16.3-22.3) | 0.55 |
| Underlying diseases, n (%) | 51 (9.7) | 12 (14) | 39 (9) | 0.26 |
| Early onset of dengue shock (<day 4), n (%) | 70 (13.4) | 23 (26) | 47 (11) | < 0.001 |
| Grade of DSS severity, n (%) | | | | < 0.001 |
| Compensated DSS | 461 (88) | 67 (75.3) | 394 (90.6) | |
| Decompensated DSS | 63 (12) | 22 (24.7) | 41 (9.4) | |
| Severe bleeding, n (%) | 55 (10.5) | 46 (51.7) | 9 (2.1) | < 0.001 |
| Systolic shock index (bpm/mmHg) | 1.3 (1.1-1.5) | 1.4 (1.3-1.7) | 1.3 (1.1-1.4) | 0.001 |
| Respiratory rate (/min) | 24 (22-28) | 30 (25-36) | 24 (20-26) | < 0.001 |
| Neurological status (on AVPU scale), n (%) | | | | < 0.001 |
| Alert and verbal response | 492 (94) | 58 (65) | 434 (99.7) | |
| Pain and unresponse | 32 (6) | 31 (35) | 1 (0.3) | |
| White blood cell count (x 10$^9$/L) | 5.19 (3.71-7.52) | 7.4 (4.6-10.8) | 4.83 (3.6-6.9) | < 0.001 |
| Hemoglobin (g/dL) | 14.8 (13.3-16.2) | 13.6 (11.6-15) | 15.1 (13.6-16.3) | < 0.001 |
| Peak hematocrit (%) | 48 (45-52) | 47 (42-51) | 48 (46-52) | 0.001 |
| Nadir hematocrit (%) | 39 (35-41) | 36 (30-41) | 39 (36-42) | < 0.001 |
| Platelet count (x 10$^9$/L) | 30 (16-49) | 29 (15-48) | 31 (16-49) | 0.64 |
| Platelet transfusion, n (%) | 88 (17) | 62 (70) | 26 (6) | < 0.001 |
| Aspartate aminotransferase (IU/L) | 148 (77-398) | 1161 (291-3158) | 125 (73-356) | < 0.001 |
| Alanine aminotransferase (IU/L) | 64 (31-202) | 412 (123-1053) | 54 (28-118) | < 0.001 |
| Severe transaminitis, n (%) | 74 (14.1) | 52 (58.4) | 22 (5.1) | < 0.001 |
| International normalized ratio (INR) | 1.28 (1.15-1.62) | 2.26 (1.77-3.0) | 1.22 (1.13-1.43) | < 0.001 |
| Serum creatinin (μmol/L) | 53 (45-61) | 57 (43-68) | 53 (45-60) | < 0.01 |
| Serum albumin (g/dL) | 2.8 (1.9-3.4) | 2.0 (1.5-2.7) | 3.1 (2.2-3.6) | < 0.001 |
| Serum lactate (mmol/L) | 2.30 (1.7-3.2) | 3.5 (2.1-6.5) | 2.2 (1.6-3.0) | < 0.001 |
| LAR | 0.86 (0.61-1.3) | 1.77 (0.91-3.38) | 0.78 (0.57-1.09) | < 0.001 |
| LB ratio | 0.14 (0.09-0.23) | 0.23 (0.13-0.4) | 0.13 (0.09-0.2) | < 0.01 |
| Arterial blood gas analysis | | | | |
| pH | 7.43 (7.39-7.47) | 7.38 (7.3-7.45) | | < 0.001 |
| PCO$_2$ (mmHg) | 27 (22-32) | 27 (22-34) | 7.44 (7.4-7.47) | 0.04 |
| PO$_2$ (mmHg) | 93 (53-139) | 139 (101-170) | 26 (22-31) | < 0.001 |
| Fraction inspired oxygen (%) | 32 (22-36) | 60 (32-60) | 81 (49-128) | < 0.001 |
| Bicarbonate (mEq/L) | 17.1 (14.7-19.4) | 16 (12.8-18.1) | 32 (21-32) 17.4 (14.8-20) | 0.11 |
| Cumulative fluid infused from referral hospitals and 24h admission, (mL/kg) | 143 (111-190) | 209 (136-303) | 137 (107-176) | < 0.001 |
| Vasoactive inotropic score within 6h of admission | 0 (0−0) | 15 (0-30) | 0 (0−0) | < 0.001 |
| *Patient complications and outcomes at discharge* | | | | |
| Mechanical ventilation requirement, n (%) | 85 (16.2) | 85 (96) | 0 (0) | < 0.001 |
| Acute liver failure, n (%) | 41 (7.8) | 41 (46) | 0 (0) | < 0.001 |
| Dengue-associated encephalitis, n (%) | 4 (0.8) | 4 (4.5) | 0 (0) | 0.001 |
| Fatal outcome, n (%) | 22 (4.2) | 22 (24.7) | 0 (0) | < 0.001 |

Summary statistics are presented as median (interquartile) for continuous variables and frequency (%) for categorical variables. Two-sided, two-sample *t* tests were used for comparison of continuous variables, and the Chi-square test (adjusted Fisher exact test) were applicable categorical group comparisons. DSS, dengue shock syndrome; LAR, Lactate-to-albumin ratio; LB, Lactate-to-bicarbonate ratio

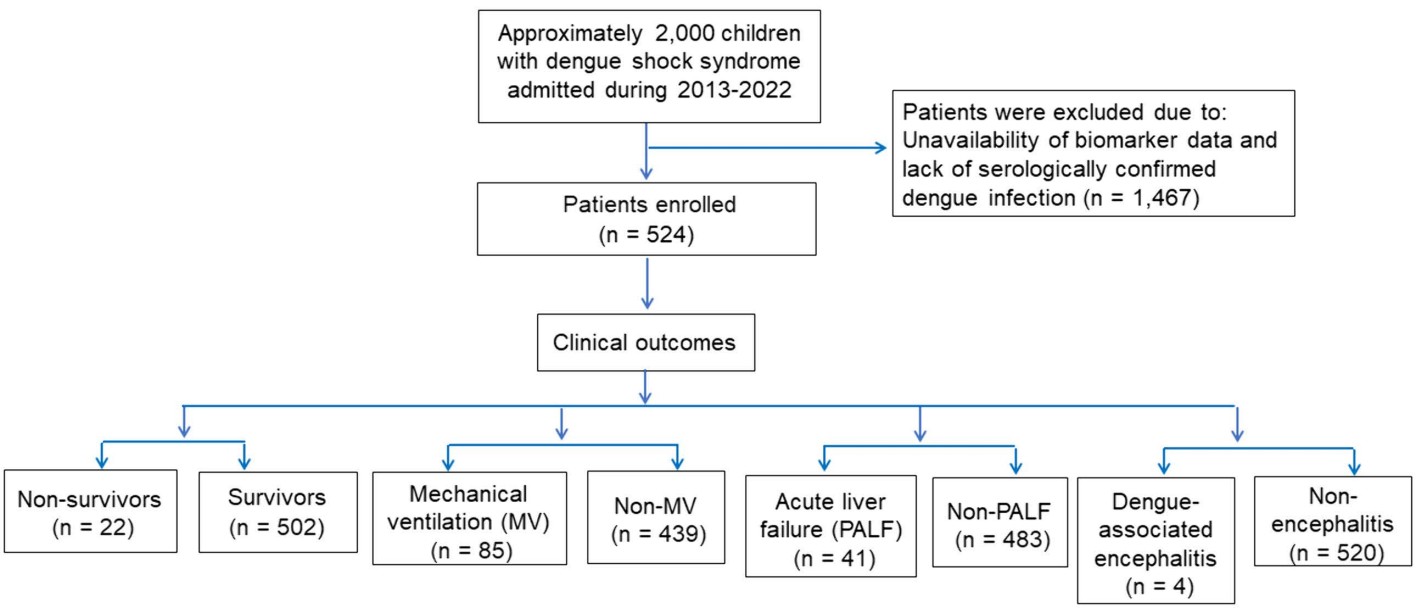

**Fig 1. The study flowchart.**

Therefore, the composite endpoints of death, MV, PALF, and dengue-associated encephalitis accounted for approximately 17% of the cohort.

### Prognostic value of lactate-based biomarkers in children with DSS

The ability of various biomarkers, including blood lactate and lactate-based ratios (LAR and LB ratio), to predict critical outcomes in children with dengue shock syndrome upon PICU admission is illustrated in Table 2. Among these, the lactate-to-albumin ratio (LAR) exhibited the strongest predictive value for critical dengue-related outcomes, achieving an area under the curve (AUC) of 0.82 (95% CI, 0.76–0.87). Serum lactate presented an AUC of 0.72 (95% CI, 0.65–0.78), while the lactate-to-bicarbonate (LB) ratio had an AUC of 0.68 (95% CI, 0.62–0.74). Notably, the blood lactate and lactate-derived ratios showed statistically significant p-values for their ability to predict composite critical dengue-associated outcomes. Based on the Youden index criteria, the optimal cutoff for LAR was 1.25, yielding a positive likelihood ratio (LR+) of 3.95, negative likelihood ratio (LR–) of 0.36, and diagnostic odds ratio of 10.97, indicating good discriminatory performance for the composite endpoint. For the LB ratio, the optimal threshold was 0.20, with LR+ of 2.26, LR– of 0.58, and diagnostic odds ratio of 3.89.

### Associations between the predefined variables and composite outcome

Unadjusted logistic regression analyses showed that most predefined covariables were significantly associated with the composite outcome (S3 Table). Female sex, presence of underlying diseases, platelet count, and serum bicarbonate were not statistically significant but were retained in the full model based on clinical and pathophysiological relevance.

### Performance of supervised models for composite endpoint in DSS patients

Table 3 summarizes the predictive performance metrics of multiple supervised machine learning models employed to identify the composite clinical endpoints in patients diagnosed with dengue shock syndrome. Logistic regression (LR)

**Table 2. Area under the receiver operating characteristic curve for serum lactate, lactate-to-albumin and lactate-to-bicarbonate ratios, and cutoff points for composite clinical endpoint among children with dengue shock syndrome.**

| Parameters | AUC of composite clinical endpoint (95% CI) | | p-value | |
|---|---|---|---|---|
| Lactate (mmol/L) | 0.72 (0.65 - 0.78) | | < 0.001 | |
| LAR | 0.82 (0.76 - 0.87) | | < 0.001 | |
| LB ratio | 0.68 (0.62 - 0.74) | | < 0.01 | |
| Cutoff points | Sensitivity | Specificity | LR (+) | LR (-) |
| LAR ≥ 1.25 | 0.71 (0.60–0.80) | 0.82 (0.78–0.86) | 3.95 (3.35–4.65) | 0.36 (0.29–0.44) |
| LB ratio ≥ 0.20 | 0.56 (0.45–0.67) | 0.75 (0.71–0.79) | 2.26 (1.84–2.79) | 0.58 (0.47–0.72) |

AUC, Area under the curve; 95% CI, Confidence interval (AUC, sensitivity, specificity, LR, Likelihood ratio-positive LR (+) and negative LR (-); LAR, Lactate-to-albumin ratio; LB, Lactate-to-bicarbonate ratio

**Table 3. Performance of supervised models to estimate the risk of the critical clinical dengue outcomes children admitted with dengue shock syndrome (N = 524).**

| Models | AUC | Sensitivity | Specificity | Precision | F1 score | Accuracy |
|---|---|---|---|---|---|---|
| LR | 0.97 (0.95- 0.99) | 0.81 (0.67-0.96) | 0.95 (0.89-0.98) | 0.85 (0.70-0.97) | 0.83 (0.71-0.93) | 0.91 (0.86-0.96) |
| RF | 0.97 (0.94-0.99) | 0.78 (0.62-0.92) | 0.96 (0.92-1) | 0.88 (0.73-1) | 0.82 (0.70-0.92) | 0.91 (0.86-0.96) |
| XGBoost | 0.96 (0.92 - 0.99) | 0.70 (0.52-0.88) | 0.94 (0.88-0.98) | 0.79 (0.63-0.94) | 0.75 (0.60-0.87) | 0.88 (0.81-0.93) |
| AdaBoost | 0.95 (0.90 - 0.98) | 0.76 (0.62 - 0.89) | 0.97 (0.93-0.99) | 0.88 (0.74-0.97) | 0.81 (0.71-0.90) | 0.92 (0.87-0.96) |
| SVM | 0.95 (0.91 - 0.98) | 0.78 (0.62-0.92) | 0.92 (0.86-0.98) | 0.78 (0.61-0.92) | 0.78 (0.65-0.89) | 0.89 (0.82-0.94) |
| KNN | 0.90 (0.83 - 0.96) | 0.62 (0.45-0.77) | 0.99 (0.97 - 1) | 0.96 (0.86 - 1) | 0.75 (0.61-0.86) | 0.91 (0.85-0.95) |
| Naïve Bayes | 0.96 (0.93 - 0.99) | 0.78 (0.62 - 0.92) | 0.92 (0.86 - 0.98) | 0.78 (0.62 - 0.92) | 0.78 (0.65 - 0.88) | 0.89 (0.82-0.94) |

AUC, Area under curve; LR, Logistic regression; RF, Random Forest; KNN, K-nearest neighbors; SVM, Support vector machine; XGBoost, eXtreme Gradient Boosting

demonstrated high discriminative ability, achieving an AUC of 0.97, with a sensitivity, specificity, precision, F1 score, and overall accuracy of 0.81, 0.95, 0.85, 0.83, and 0.91, respectively. Despite this strong performance, the LR model may exhibit overfitting, as evidenced by skewed distributions of several clinical and biochemical features (S2 Fig, S2 Table). Therefore, ensemble and kernel-based models were analyzed to mitigate this limitation and improve generalizability.

Among the supervised models, random forest, AdaBoost, XGBoost, and the support vector machine consistently yielded excellent predictive results across all evaluation metrics. The RF model demonstrated strong predictive performance, achieving an AUC of 0.97, with a sensitivity of 0.78, specificity of 0.96, precision of 0.88, F1 score of 0.82, and accuracy of 0.91. The SVM model also performed robustly, with an AUC of 0.95, sensitivity of 0.78, specificity of 0.92, precision of 0.78, F1 score of 0.78, and an accuracy of 0.89. Both AdaBoost and XGBoost exhibited strong classification capabilities, although with marginally reduced sensitivity. Additionally, other supervised learning algorithms, such as k-nearest neighbors and Naïve Bayes, have demonstrated competitive performance, supporting their potential utility in predicting adverse outcomes in patients with DSS.

## Important predictive variables identified by random forest model

The random forest model was utilized to determine the most influential clinical features and biomarkers associated with the prediction of the composite clinical endpoint in patients with dengue shock syndrome, as illustrated in Fig 2. Among the evaluated variables, the LAR emerged as the most informative biomarker, outperforming both the serum lactate concentration and the lactate-to-bicarbonate ratio in terms of predictive accuracy. This highlights the enhanced discriminative value of LAR in identifying dengue-infected patients at risk of critical clinical outcomes. In addition to LAR, the RF model also identified several key clinical parameters contributing significantly to model performance. These included indicators of disease severity and hemodynamic compromise, such as severe bleeding, elevated INR, platelet transfusion requirement, cumulative volume of fluid resuscitation administered both at the referring hospital and within the first 24h of PICU admission, severe transaminitis, deterioration in neurological status as assessed by AVPU scale, a high vasoactive-inotropic score (> 30), increased respiratory rate, elevated serum creatinine levels, and an increased systolic shock index. Collectively, these variables reflect the multifactorial pathophysiology of DSS and underscore the utility of ensemble models in identifying the complex interactions among clinical predictors.

## Interpretation of predictive features using SHAP analysis

The Shapley Additive Explanations analysis derived from the ensemble model (XGBoost) identified and ranked the most influential variables associated with critical clinical outcomes in hospitalized pediatric patients with DSS, as presented in Fig 3. The SHAP summary plot demonstrated that all prespecified clinical predictors and biomarkers contributed substantially to the model's output, reinforcing their clinical relevance in this high-risk population. Among these, INR exhibited the

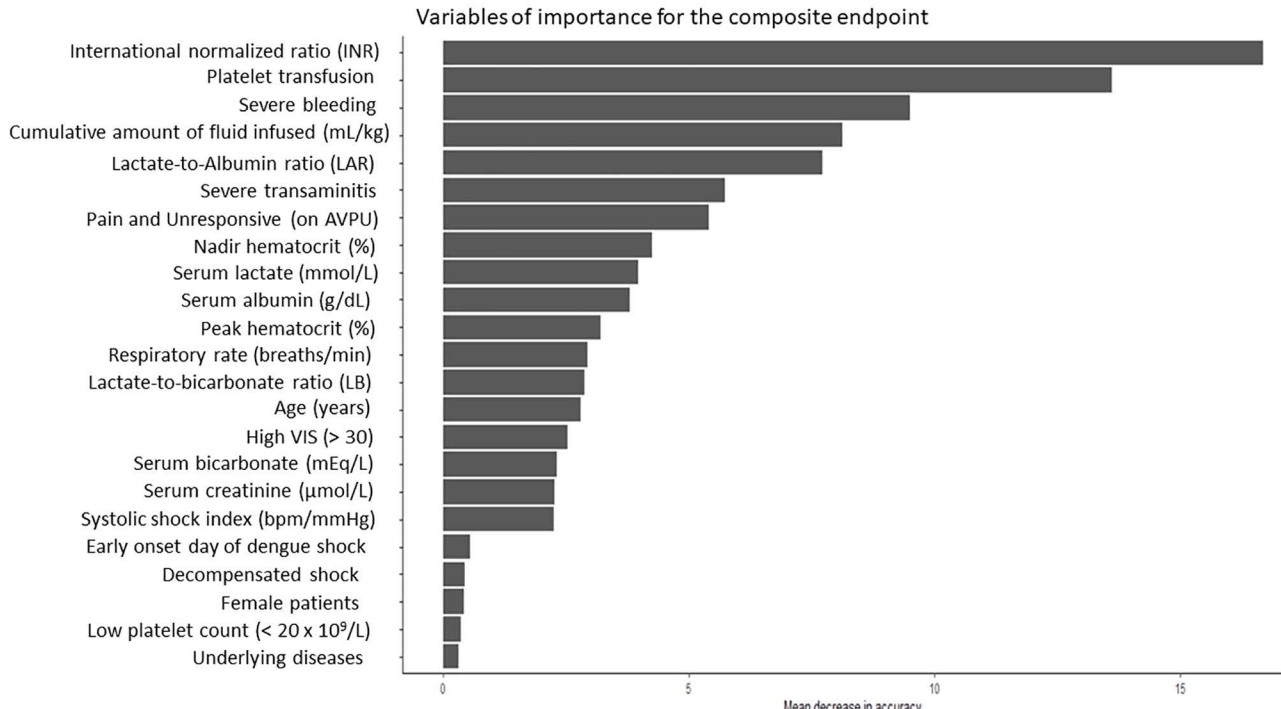

**Fig 2. Key predictors of the composite clinical endpoint in dengue shock syndrome identified by the random forest model.** The lactate-to-albumin ratio (LAR) was the strongest predictor, outperforming serum lactate and the lactate-to-bicarbonate (LB) ratio. Additional important variables included markers of hemodynamic compromise, organ dysfunction, and disease severity, reflecting the multifactorial pathophysiology of dengue shock syndrome.

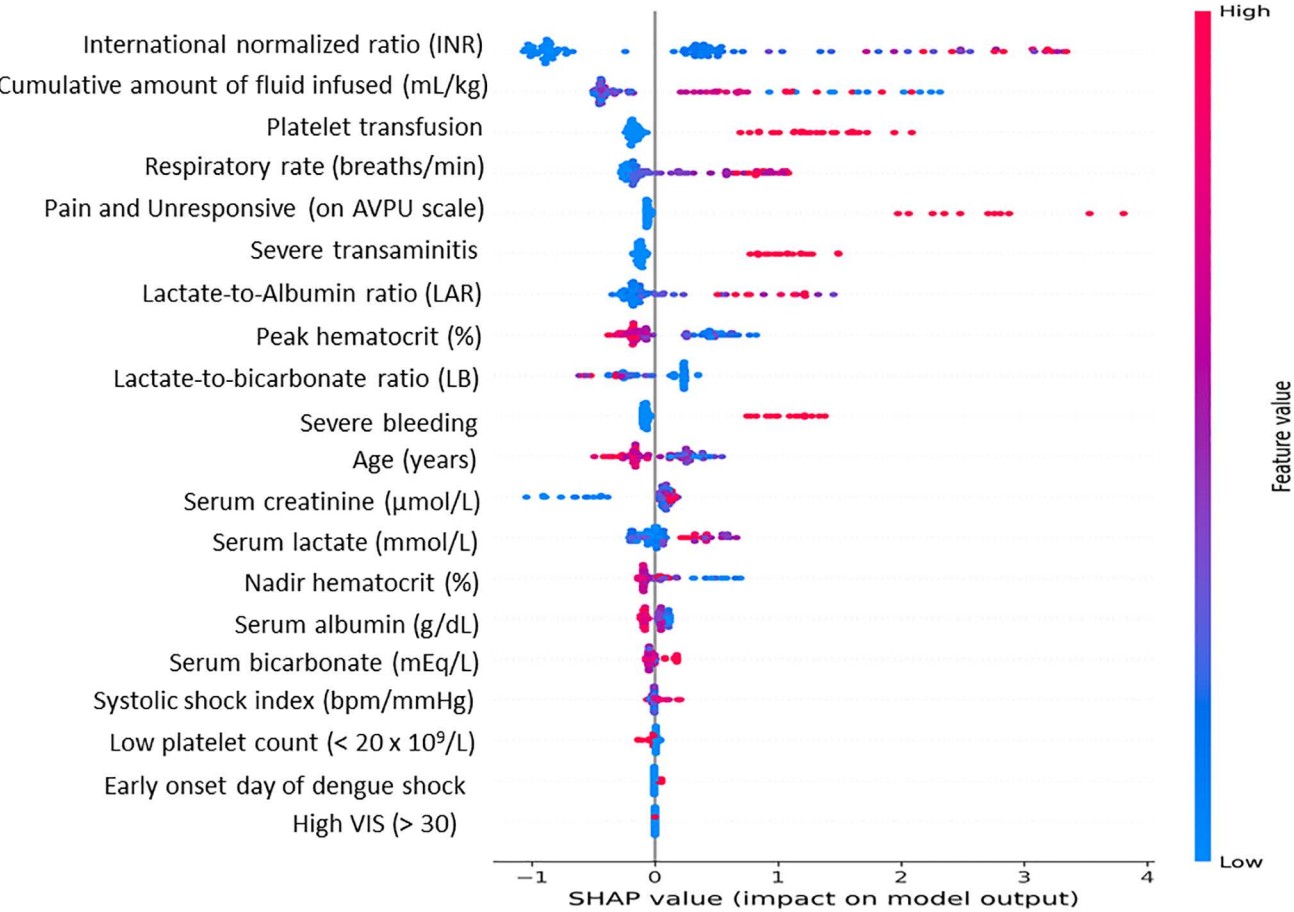

**Fig 3. SHAP summary plot of predictors of critical outcomes in pediatric dengue shock syndrome.** SHAP analysis identified elevated INR, platelet transfusion requirement, high fluid administration, and respiratory failure as the strongest clinical predictors, whereas LAR was the most informative biomarker, followed by LB ratio and serum lactate. These results emphasize the prognostic utility of serum lactate-derived indices and the importance of multiple clinical features in determining the outcomes.

highest SHAP values, indicating that it is a prominent predictor of critical dengue-related outcomes. Furthermore, a higher cumulative amount of intravenous fluid administration, both prior to admission from referring hospitals and during the first 24 h of PICU admission, along with the presence of respiratory failure, was strongly associated with the study endpoint of the requirement for mechanical ventilation.

Among the biomarkers, LAR had the highest SHAP value, followed by the LB ratio and lactate, highlighting the prognostic superiority of lactate-derived indices. In contrast, single biomarkers, such as serum albumin and bicarbonate, contributed modestly to the overall model prediction. Reduced consciousness (on AVPU scale), severe bleeding, and the requirement for platelet transfusion were strong predictors of critical outcomes. Additional key features identified included indicators of hemodynamic instability and disease severity, such as an elevated systolic shock index and high vasoactive-inotropic score, which reflect both circulatory compromise and the intensity of pharmacological cardiovascular support. Other important contributors included younger age, severe transaminitis, profound thrombocytopenia (platelet count < 20 × 10⁹/L), hematocrit (%), and increased serum creatinine levels, all of which were strongly associated with the development of severe disease manifestations in children with DSS.

## Performance of predictive models, internal validation and calibration plots

Compared with the LAR- and LB-based models, the lactate-based predictive model demonstrated a lower Brier score (0.03 in training set and 0.04 in testing set), higher c-statistics and fairly good calibration slope (Table 4). Nevertheless, all three predictive models exhibited largely comparable c-statistics and Brier scores, and overall, each showed reasonably good calibration performance. Calibration plots were generated for the LAR model (Fig 4A), Lactate model (Fig 4B), and LB model (Fig 4C). The lactate-based prognostic model for the composite dengue endpoint demonstrated good agreement between the predicted probabilities and observed outcomes. In contrast, the LAR and LB models showed acceptable calibration between the predicted and observed values.

**Table 4. Performance of predictive models with internal validation.**

| Models | C-statistic | Calibration-in-the-large | Calibration slope | Brier score |
|---|---|---|---|---|
| LAR | 0.96 (0.93) | 0 (−0.28) | 1 (0.72) | 0.03 (0.04) |
| Lactate | 0.96 (0.93) | 0 (−0.27) | 1 (0.74) | 0.03 (0.04) |
| LB | 0.95 (0.93) | 0 (−0.25) | 1 (0.76) | 0.04 (0.05) |

Statistics are summarized for training (test) sets performed with bootstrap (n = 500 resampling) and internal validation. LAR, Lactate-to-albumin ratio; LB, Lactate-to-bicarbonate ratio

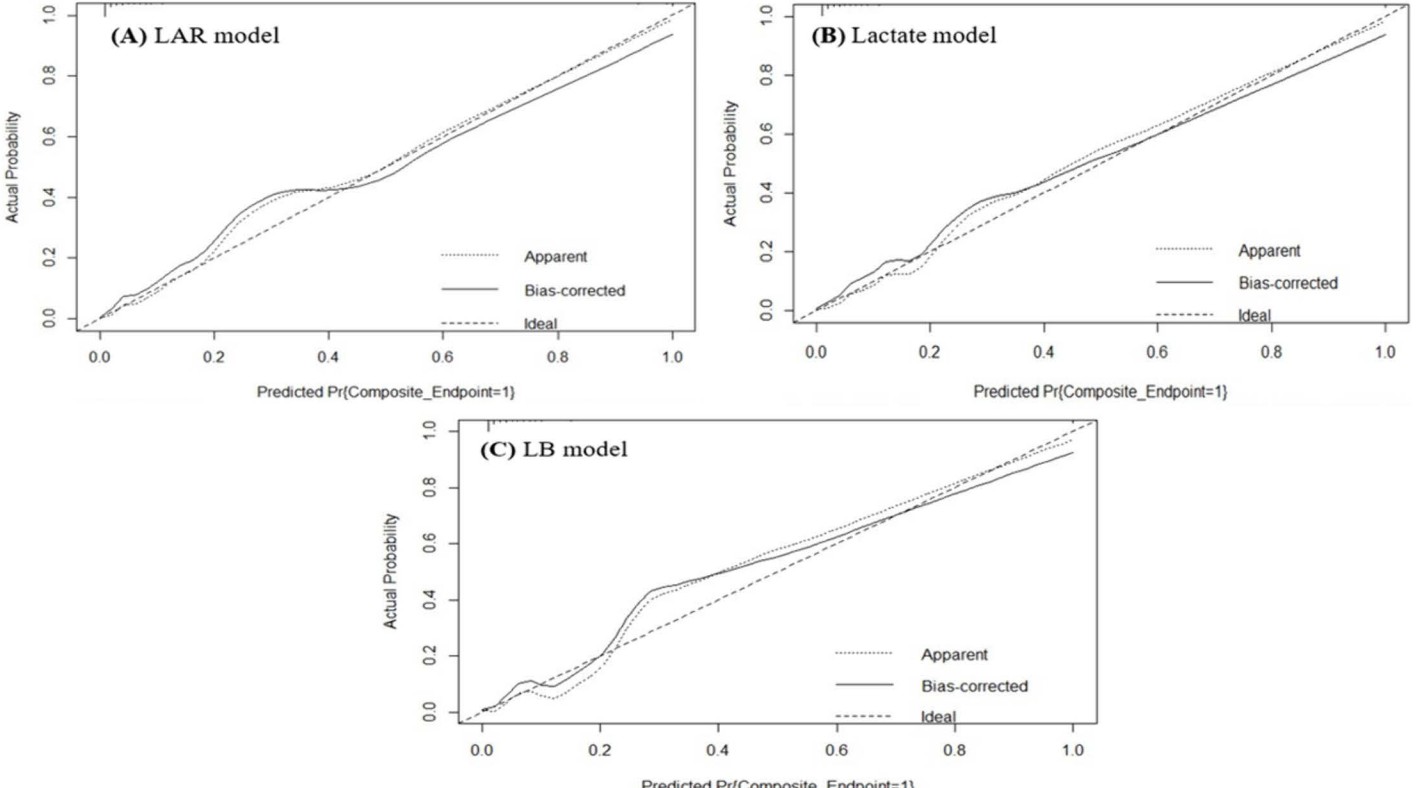

**Fig 4. Calibration plots for prognostic models.** Calibration plots are presented for the (A) LAR (B) Lactate, and (C) LB models.

## Decision curve analysis

As shown in **Fig 5**, the decision curve analysis indicates that the LAR, LB, and lactate-based models may provide a net clinical benefit across a wide range of threshold probabilities. Among these, the LAR-based model demonstrated the greatest net benefit compared with the LB and lactate models. These results suggest that these biomarkers may have potential clinical utility.

## Discussion

Pediatric patients with dengue shock syndrome who develop severe complications, such as respiratory failure requiring mechanical ventilation, dengue-associated pediatric acute liver failure (PALF), or encephalitis, have a markedly elevated risk of mortality, reportedly ranging from 20% to 60% [2,15,16]. While individual biomarkers, such as serum lactate, albumin, or bicarbonate, are commonly used in clinical practice, each alone may insufficiently capture the multifactorial nature of DSS pathophysiology. Therefore, combining these markers into derived indices, specifically the lactate-to-albumin ratio (LAR) and the lactate-to-bicarbonate ratio (LB), may enhance prognostic accuracy. However, data on the utility of the LAR and LB ratio in predicting critical complications in pediatric DSS populations remain limited. The findings from this study highlight the prognostic value of the composite biomarkers, supporting their potential role in improving early risk stratification, guiding clinical decision-making, and ultimately enhancing survival outcomes in pediatric dengue care.

To date, many studies have identified elevated serum lactate concentrations and persistent hyperlactatemia as important prognostic markers associated with an increased mortality risk in PICU-admitted patients. Additionally, the lactate-to-albumin ratio (LAR) has emerged as a valuable biomarker for predicting in-hospital mortality in critically ill patients [17]. Elevated LAR has been consistently associated with increased 28-day mortality rates in both general intensive care and sepsis-specific cohorts, underscoring its potential utility in risk stratification and clinical outcome prediction [17–20]. Notably, recent findings from our research group indicate that the lactate-to-bicarbonate (LB) ratio exhibits marginally superior prognostic performance compared with serum lactate alone in predicting 28-day mortality among pediatric patients with DSS, supporting its potential clinical utility as a prognostic biomarker [4]. To date, no studies have specifically examined

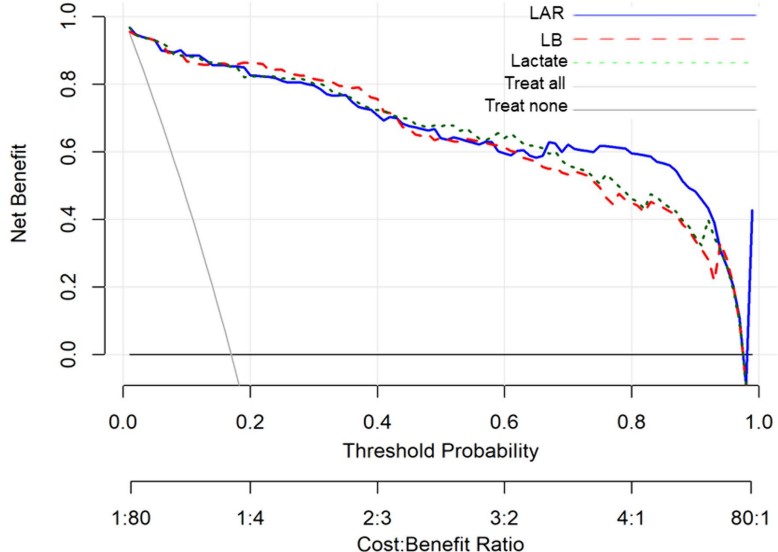

**Fig 5. Decision curve analysis of prognostic models.** The LAR-, LB-, and lactate-based models showed a net clinical benefit across a range of threshold probabilities, with the LAR model demonstrating the highest benefit.

the prognostic value of the lactate-to-albumin ratio in pediatric patients with dengue shock syndrome. Moreover, comparative evaluations against the LB ratio or individual biomarkers, such as lactate, are lacking. Thus, our study has been conducted to address this research gap.

In this study, 17% of the cohort had composite critical endpoints (death, MV, dengue-associated PALF, and encephalitis). These figures are consistent with those of previous cohort studies [21,22]. Notably, we incorporated a composite outcome encompassing severe clinical complications to enhance statistical power, given the relatively low mortality rate (4.2%) observed in the dengue shock syndrome cohort. Receiver operating characteristic curve analysis demonstrated that LAR exhibited superior discriminative performance, with an area under the curve (AUC) of 0.82, compared to serum lactate (AUC 0.72) and the lactate-to-bicarbonate (LB) ratio (AUC 0.68) in predicting critical dengue-related outcomes. These findings align with those of prior studies that investigated the prognostic value of LAR in critically ill adult patients [19,20]. Nevertheless, our study is among the first to extend this application to pediatric patients with dengue shock syndrome. Notably, the optimal cutoff value for LAR in our pediatric cohort was 1.25, yielding a sensitivity of 71%, specificity of 82%, and an overall accuracy of 80%. This is comparable to previous findings in adult patients with sepsis, where Bou Chebl *et al.* reported cutoffs of 1.22 in sepsis and 1.47 in septic shock with lower predictive values [20]. For the LB ratio at PICU admission, a threshold of 0.20 provided moderate predictive utility (sensitivity 56%, specificity 76%, accuracy 72%). In contrast, our previous work showed that the LB ratio measured within 24 h of PICU admission had greater prognostic accuracy for 28-day in-hospital mortality, with a cutoff of 0.25 yielding a sensitivity of 73%, specificity of 83%, and accuracy of 83% [4]. These findings suggest that lactate-based ratios measured later at 24h within the PICU may better capture the evolving pathophysiological deterioration associated with DSS. Nevertheless, the original dataset retrieved lacked 24 h post-admission albumin data, so that at-24h LAR was impeded from analysis.

Our recent publications have identified a range of key clinical variables significantly associated with critical outcomes in patients with dengue shock syndrome (DSS), including younger age, female sex, early onset of DSS (prior to day 4 of illness), presence of comorbidities, severe bleeding, marked transaminitis, profound thrombocytopenia (< $20 \times 10^9$/L), requirement for platelet transfusion, elevated international normalized ratio (INR), decompensated shock, hemodynamic instability, large cumulative fluid resuscitation volume, elevated vasoactive-inotropic score, altered consciousness (as assessed by the AVPU scale), respiratory distress, and acute kidney injury [21,22]. Given the expanding role of machine learning in predictive modeling across diverse clinical domains, our study is among the first to apply supervised machine learning algorithms to evaluate the prognostic performance of lactate-derived indices, namely the lactate-to-albumin ratio (LAR) and lactate-to-bicarbonate ratio (LB), in conjunction with established clinical predictors in pediatric DSS. Although logistic regression yielded a strong predictive performance, evidence of potential overfitting was noted, as indicated by skewed feature distributions across clinical and laboratory variables. To mitigate this and enhance generalizability, we employed ensemble models (RF, XGBoost and AdaBoost) and kernel-based methods (SVM and KNN). The LAR consistently outperformed the LB ratio and individual biomarkers, such as serum lactate, albumin, and bicarbonate. This finding was further supported by the results from supervised models, particularly ensemble learning analyses, which have shown strong performance in biomedical research by reducing overfitting and improving generalizability across nonlinear and high-dimensional datasets [23]. To substantiate the model interpretability, SHAP analysis was conducted, which revealed that LAR had the highest feature importance, followed by the LB ratio and serum lactate. In contrast, serum albumin and bicarbonate alone contributed minimally to the overall model prediction, reinforcing the superior prognostic value of LAR compared to other lactate-based and individual biomarkers.

The superior prognostic performance of the lactate-to-albumin ratio (LAR) underscores its relevance in capturing the complex and interrelated pathophysiological processes underlying severe dengue shock syndrome. As the denominator of the ratio, serum albumin serves as a surrogate marker for the severity of plasma leakage, which is the hallmark of dengue pathogenesis. Given its biological half-life of approximately three weeks [24], a rapid and profound decline in albumin levels in the acute phase of DSS cannot be attributed solely to impaired hepatic synthesis or increased catabolism. Rather,

this decline reflects the substantial redistribution of albumin secondary to dengue-induced endothelial injury. This process may be further compounded by hemodilution resulting from aggressive fluid resuscitation, which is associated with adverse clinical outcomes [22,25]. Furthermore, serum lactate, the numerator of the LAR, indicates the metabolic consequences of plasma leakage, including impaired perfusion, anaerobic metabolism, and systemic circulatory failure. In DSS, elevated lactate levels not only signal the presence of shock but also reflect inadequate tissue oxygenation despite fluid resuscitation and vasopressor support. Therefore, LAR captures both the underlying pathophysiological insult (plasma leakage resulting in hypoalbuminemia) and its clinical consequences (refractory shock, evidenced by hyperlactatemia). This integrated metric provides a more comprehensive representation of dengue severity than either lactate or albumin alone. High LAR values may indicate excessive capillary leakage or evolving multiorgan failure, particularly in patients with dengue-associated acute liver failure who present late in the course of irreversible shock. Accordingly, the capacity of LAR to quantify the interplay between vascular pathology and systemic perfusion failure likely underlies its superior prognostic value in predicting critical outcomes in pediatric DSS patients. In addition, the relatively modest prognostic performance of the LB ratio may be explained by its high susceptibility to physiological and clinical variability [4]. While lactate levels rise consistently in response to tissue hypoperfusion, serum bicarbonate is more labile, reflecting not only metabolic acidosis, but also compensatory mechanisms, fluid resuscitation, and renal handling of acid–base balance [26]. This variability may dilute the predictive signal of the LB ratio, making it less stable than serum lactate and LAR. Nevertheless, LAR integrates lactate with albumin, a biomarker less influenced by short-term metabolic fluctuations, thereby offering a more robust reflection of both metabolic stress and plasma leakage in patients with severe dengue.

On admission, LAR appears to be an informative early marker of poor outcomes in children with DSS, capturing critical pathophysiological processes that extend beyond hyperlactatemia alone. While elevated lactate levels reflect impaired perfusion and anaerobic metabolism, concurrent hypoalbuminemia at admission likely reflects the combined effects of systemic inflammation, capillary leakage, and underlying nutritional status. This broader biological context may explain the higher discriminatory performance of LAR compared to serum lactate or LB ratio, particularly at the time of hospital presentation. Most significantly, it should be acknowledged that LAR, LB, and lactate values may evolve over the first 24 h owing to fluid resuscitation, hemodilution, and ongoing vascular leakage, which could modify their predictive utility over time after hospitalization [22,25]. Notably, our recent report showed a higher predictive value of lactate level and LB ratio at 24h post-admission than at the time of admission in this study [4]. Therefore, this suggests the need for dynamic modelling of these biomarkers during the first 24h of PICU admission. An additional consideration is that prognostication against a heterogeneous composite endpoint may obscure the differential predictive performance for specific complications, such as dengue-associated acute liver failure or the need for mechanical ventilation. These findings support the potential clinical relevance of LAR, highlighting the need for further validation across settings and outcome definitions. Another significant point is that the decision curve analysis incorporating admission LAR into early risk stratification may yield actionable benefits, such as timely intensive care escalation, prioritizing intensified hemodynamic monitoring, and prompting early consultation for patients at greatest risk.

This study had several strengths. The predictive models were developed using a clinically relevant cohort of hospitalized pediatric patients with dengue shock syndrome and severe dengue, encompassing a broad spectrum of critical clinical outcomes. Key predictive features were determined by integrating comprehensive clinical expertise with robust machine-learning methodologies, thereby improving the reliability and interpretability of the findings. The study findings offer substantial clinical value, particularly in facilitating early risk stratification, supporting informed clinical decision-making, and potentially improving survival outcomes in the management of pediatric dengue. However, this study had certain limitations. This retrospective, single-center study may constrain the generalizability of our findings. Additionally, variability in the timing and standardization of clinical and laboratory data collection at the time of hospital admission may have introduced information bias. It is notable for the risk of overfitting without nested or temporal validation and the potential bias introduced by excluding variables with > 20% missingness. As this study was based on secondary data

analysis, it was constrained by the scope and quality of the information collected in the original investigation, with no possibility of verifying or supplementing missing variables. Furthermore, the consistency of the recorded data may have been affected by variations in clinical practice and documentation during the study period. A further limitation was the relatively low number of mortality events, which may have led to outcome imbalance; however, this was addressed by constructing a composite endpoint to increase the statistical power and ensure meaningful outcome assessment. Additionally, the distribution of predictor variables exhibited skewness, which posed a risk of overfitting in models such as logistic regression. To mitigate this, ensemble methods were employed, which demonstrated greater robustness in handling non-normal feature distributions and class imbalance.

This study highlights the potential of lactate-derived biomarkers, particularly the lactate-to-albumin ratio, as practical tools for early risk stratification in pediatric dengue shock syndrome. Compared with the LB ratio and serum lactate alone, LAR demonstrated superior prognostic performance in predicting critical complications, including mechanical ventilation, acute liver failure, and dengue-associated encephalitis. Importantly, both serum lactate and albumin tests are routinely available and low-cost laboratory measurements in most provincial and district hospitals, making the LAR feasible for implementation, even in resource-limited settings. Incorporating LAR into admission triage protocols may allow for the earlier identification of high-risk children, support timely referral to higher-level facilities, and guide the prioritization of intensive monitoring and therapeutic interventions. Nevertheless, prospective validation across diverse healthcare systems is needed to confirm its generalizability before it can be adopted as a standard risk assessment tool.

## Conclusion

The lactate-to-albumin ratio demonstrated superior predictive performance compared to serum lactate or the lactate-to-bicarbonate ratio in identifying in-hospital complications, such as mortality, the need for mechanical ventilation, acute liver failure, and encephalitis. These findings underscore the clinical utility of LAR as a feasible and readily available biomarker for risk stratification in the management of pediatric dengue patients.

## Supporting information

**S1 Table.  Variables used in the supervised machine learning models.**
(DOCX)

**S2 Table.  Standardized mean differences (SMD) for covariables assessing distributional balance between outcome groups.**
(DOCX)

**S3 Table.  The unadjusted associations between the covariables and composite endpoint.**
(DOCX)

**S1 Fig.  Missing value chart of candidate variables.**
(TIF)

**S2 Fig.  Distributions of clinical and biomarker variables in machine learning models.**
(TIF)

**S1 File.  The least absolute shrinkage and selection operator method.**
(DOCX)

**S2 File.  A study analysis flowchart for predictive model development.**
(DOCX)

**S1 Appendix.  Observational data of 524 participants in this study cohort.**
(XLSX)

**S2 Appendix.  All statistical analysis coding with R (version 4.5.0) and Python (Anaconda Distribution version 3.0).**
(DOCX)

**S3 Appendix.  Complete observational data of all participants from the primary research.**
(XLSX)

**S1 Checklist.  TRIPOD checklist.**
(DOCX)

## Acknowledgments

We are grateful to the patients, administrative staffs, Dr. Tuong TTH and Mr. Cuong NH for their participation in this study.

## Author contributions

**Conceptualization:** Nguyen Tat Thanh , Vo Thanh Luan.

**Data curation:** Nguyen Tat Thanh , Vo Thanh Luan.

**Formal analysis:** Nguyen Tat Thanh .

**Funding acquisition:** Vo Thanh Luan.

**Investigation:** Nguyen Tat Thanh , Vo Thanh Luan.

**Methodology:** Nguyen Tat Thanh , Vo Thanh Luan.

**Project administration:** Vo Thanh Luan.

**Resources:** Vo Thanh Luan.

**Supervision:** Nguyen Tat Thanh , Vo Thanh Luan.

**Writing – original draft:** Nguyen Tat Thanh , Vo Thanh Luan.

**Writing – review & editing:** Nguyen Tat Thanh , Vo Thanh Luan.

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
