## [Decision Letter · Decision Letter 0]

15 Aug 2025

Dear Dr. Thanh ,

Thank you for submitting your manuscript to PLOS ONE. After careful consideration, we feel that it has merit but does not fully meet PLOS ONE’s publication criteria as it currently stands. Therefore, we invite you to submit a revised version of the manuscript that addresses the points raised during the review process.

We look forward to receiving your revised manuscript.

Kind regards,

Muhammad Iqhrammullah, Ph.D

Academic Editor

PLOS ONE

Journal Requirements:

3. We note that there is identifying data in the Supporting Information file <S1 Appendix.xlsx>. Due to the inclusion of these potentially identifying data, we have removed this file from your file inventory. Prior to sharing human research participant data, authors should consult with an ethics committee to ensure data are shared in accordance with participant consent and all applicable local laws.

-Location data

Please remove or anonymize all personal information (<ID and Age>), ensure that the data shared are in accordance with participant consent, and re-upload a fully anonymized data set. Please note that spreadsheet columns with personal information must be removed and not hidden as all hidden columns will appear in the published file.

Reviewers' comments:

Reviewer's Responses to Questions

**Comments to the Author**

1. Is the manuscript technically sound, and do the data support the conclusions?

Reviewer #1: Partly

Reviewer #2: Yes

Reviewer #3: Yes

Reviewer #4: Yes

2. Has the statistical analysis been performed appropriately and rigorously?

Reviewer #1: Yes

Reviewer #2: Yes

Reviewer #3: Yes

Reviewer #4: Yes

3. Have the authors made all data underlying the findings in their manuscript fully available?

Reviewer #1: No

Reviewer #2: Yes

Reviewer #3: Yes

Reviewer #4: Yes

4. Is the manuscript presented in an intelligible fashion and written in standard English?

Reviewer #1: Yes

Reviewer #2: Yes

Reviewer #3: Yes

Reviewer #4: Yes

Reviewer #1: Title & Abstract — Please temper superlatives such as “excellent performance” unless supported by external validation and demonstrated clinical utility; present all performance metrics with 95% confidence intervals using consistent precision (for example, AUC 0.82, 95% CI 0.78–0.86) and include PR-AUC given class imbalance; explicitly list the composite outcome components and total event count in the abstract; and state the rule used to choose the “optimal” LAR cutoff (e.g., Youden index), reporting sensitivity and specificity with 95% CIs at that threshold.

Introduction — Clarify the biological and clinical rationale for why LAR may outperform lactate or L/B in DSS, briefly contrasting prior 24-hour findings with admission-time measurements; define each component of the composite endpoint (death, mechanical ventilation, PALF, encephalitis) and justify their aggregation from a clinical-decision perspective; and close with a precise objective that commits to assessing both discrimination and calibration while comparing LAR against lactate and L/B.

Methods: Study Design & Cohort — Add a transparent flow diagram detailing numbers screened, excluded with reasons, and included; specify inclusion criteria, timing of blood sampling relative to admission, handling of any repeated measures, and follow-up window; provide exact outcome definitions (including whether mechanical ventilation includes non-invasive ventilation) and per-component counts; and state whether a protocol existed or confirm that no prior registration was performed.

Methods: Predictors & Missing Data — Provide a complete list of candidate predictors with units and admission-time availability and include a data dictionary as supplementary material; characterize missingness at the variable and patient levels and discuss the likely mechanism (MAR/MNAR), supported by a missingness heatmap; perform imputation within the resampling loop so that models only “see” imputed values created from their training folds; justify the >20% missingness exclusion rule, list excluded variables, and present a complete-case sensitivity analysis.

Methods: Preprocessing & Modeling — Ensure that all preprocessing (scaling, one-hot encoding, class weighting or resampling, and any feature selection) is fit exclusively on training folds and then applied to held-out data; replace the single split with nested cross-validation or bootstrap optimism correction and, if feasible, add a temporal split (earlier years to train, later years to test) to probe transportability; report AUC, PR-AUC, Brier score, and expected calibration error with 95% CIs; include calibration-in-the-large and slope, add calibration plots, and consider isotonic or Platt recalibration; present decision-curve analysis to compare net benefit against treat-all/none policies and simple heuristics such as lactate alone; and show threshold-level confusion matrices, sensitivity, specificity, PPV, NPV, and likelihood ratios at clinically relevant cut points, specifying statistical tests (e.g., DeLong) and any multiplicity adjustments.

Results: Cohort & Outcomes — Provide a baseline characteristics table stratified by outcome with standardized mean differences to convey distributional separation; report per-component outcome counts and, where possible, univariable associations of LAR, lactate, and L/B with each component; when proposing cutoffs, give sensitivity/specificity with CIs and LR+/LR− alongside prevalence-adjusted PPV/NPV to guide bedside interpretation; and for machine-learning models, present outer-fold performance (mean ± SD), calibration results, and decision-curve plots.

Results: Comparative Performance — When stating that LAR outperformed lactate and L/B, quantify the difference using ΔAUC with 95% CIs and p-values, and corroborate with PR-AUC due to class imbalance; probe robustness with complete-case analyses and, if available, a temporal split; and assess whether apparent gains are driven largely by one component of the composite by providing per-component performance or sensitivity analyses that exclude a dominant component.

Discussion — Interpret the findings with calibrated language emphasizing that admission LAR is an informative early marker while acknowledging that later (e.g., 24-hour) measurements may differ; relate LAR’s performance to plausible DSS pathophysiology in which albumin reflects inflammatory, nutritional, and capillary-leak states beyond lactate alone; discuss the implications and potential drawbacks of predicting a heterogeneous composite versus component-specific endpoints; and translate decision-curve results into actionable scenarios such as ICU escalation, intensified monitoring, or early hepatology consultation.

Limitations — Expand the section to note risks of overfitting without nested or temporal validation, the potential bias introduced by excluding variables with >20% missingness, the clinical heterogeneity within the composite endpoint, the single-center design with evolving practice patterns over 2013–2022, and measurement variability in albumin/lactate assays; explicitly state that external validation is needed before clinical deployment.

Figures, Tables, and Supplement — Make figure captions fully self-contained by defining all acronyms and describing each panel; ensure clear axes, units, and readable fonts with consistent significant digits; add figures for the patient flow, missingness structure, calibration curves, decision-curve analysis, SHAP explainability, and threshold-level confusion matrices; and include a supplement with full model specifications, hyperparameter grids, per-fold results, software and exact package versions, and reproducible scripts or notebooks.

Data Availability & Reproducibility — Revise the Data Availability Statement to meet PLOS ONE’s policy by depositing a de-identified dataset with an accompanying data dictionary and the full analysis code in a public repository (e.g., OSF or Zenodo with a DOI, and code mirrored on GitHub with a release archived to Zenodo); if any restrictions apply, specify them precisely and provide a compliant access mechanism; and document random seeds, software versions, and hardware used to ensure reproducibility.

Ethics & Governance — Provide the institutional review board approval identifier and date, clarify consent or waiver status, and summarize de-identification and secure data handling procedures; disclose funding sources and any conflicts of interest to ensure transparency regarding potential influences on study design, analysis, or interpretation.

Language & Style — Edit for clarity and standard English, correcting typographical issues such as “Shapley Addictive Explanations” to “Shapley Additive Explanations” and “admisison” to “admission”; define all abbreviations at first mention and use them consistently thereafter (DSS, LAR, L/B, MV, PALF, SHAP, RF, LR, SVM); standardize numerical reporting (e.g., p = 0.003 or p < 0.001, consistent AUC precision, en-dashes for ranges, n/N [%]); and shorten long sentences in the Methods and Discussion to improve readability.

Reviewer #2: Thank you for the chance to review a manuscript title “Comparison of serum lactate and lactate-derived ratios as prognostic biomarkers in pediatric dengue shock syndrome using supervised machine learning models” by Nguyen Tat Thanh and Vo Thanh Luan. I believe this work is valuable since lactate and related ratios are biochemically linked to the core pathophysiology of dengue shock syndrome in dengue infection. I have several suggestions that can be considered further by the authors; please find them below:

1. Per peer review best practice for title formatting, please avoid abbreviations in the title would enhance clarity for multidisciplinary readers—e.g., write “dengue shock syndrome” in full and avoid “ML” abbreviation.

2. In the abstract, lease explicitly address limitations or potential biases, which is crucial for biomarker performance claims.

3. To increase the reproducibility, it is better to explicitly mention parameter tuning details, data preprocessing steps, and handling of missing data (e.g., imputation method).

4 Add some potential bias considerations, such as being single-center and retrospective design.

5. Kindly avoid overstating non-significant findings (no misleading p>0.05 claims).

6. The fact that L/B ratio underperformed may relate to its higher susceptibility to metabolic compensation and bicarbonate variability, but this reasoning is not discussed.

7. Calibration assessment (e.g., Hosmer-Lemeshow, Brier score) can be done and reported for prognostic model trustworthiness.

8.Authors indeed have described LAR as “practical and accessible” but does not discuss feasibility in resource-limited settings. In many such contexts, albumin/bicarbonate assays and rapid turnaround are unavailable, and implementing complex ML models (e.g., RF, SVM) at the point of care is unrealistic without substantial infrastructure. Please narratively analyze the feasibility of implementing the assessment, especially in resource constraint setting.

Reviewer #3: This manuscript addresses an important and timely clinical question regarding the prognostic utility of lactate-derived indices. However, there are several areas that require clarification, methodological strengthening, and refinement of presentation to ensure the work meets the standards. Below are detailed comments intended to help the authors enhance the scientific robustness

1. Please elaborate on how this work differs from previous publications, particularly the 2024 Medicine (Baltimore) study on the L/B ratio. While the Discussion notes that this is among the first pediatric DSS studies comparing LAR and L/B ratio with ML, stating this clearly in the Introduction would highlight the unique contribution.

2. The reported AUCs are high (up to 0.97), which raises concerns about overfitting. It would strengthen the manuscript to:

- Clarify whether nested cross-validation was used for hyperparameter tuning, and whether outcome stratification was applied to balance rare events.

- Add calibration metrics (e.g., Brier score, calibration plots) to complement discrimination measures.

- Describe steps taken to avoid data leakage during preprocessing (e.g., imputation and scaling performed within CV folds).

3. The composite outcome (mortality, MV, PALF, encephalitis) is clinically meaningful, but the rationale for combining these different endpoints should be explained in more detail, as their prognostic weight and pathophysiology differ. If feasible, a sensitivity analysis using mortality alone or analyzing each component separately would help confirm robustness.

4. The lack of 24-hour post-admission albumin data is acknowledged, but the discussion could be expanded to consider how serial lactate and albumin measurements might improve prognostic accuracy, referencing existing literature where available.

5. In Table 3, please include 95% confidence intervals for sensitivity, specificity, and accuracy, similar to those provided for AUC. This would improve the statistical transparency of the results.

6. A brief discussion on applicability in non-tertiary or different regional settings would help contextualize the findings. In addition, note the potential selection bias introduced by excluding patients with >20% missing data.

Reviewer #4: Major Comments:

1. Consider including external validation or additional internal validation (e.g., bootstrap) to reduce the risk of overfitting.

2. Clarify the method for determining optimal cut-offs for LAR and L/B ratio.

3. Discuss the potential utility of these biomarkers in earlier clinical settings (e.g., ER triage) beyond PICU admission.

4. Provide more detail on missing data patterns and imputation procedures, including sensitivity analyses comparing imputed vs. complete cases.

5. Expand on the feasibility and cost-effectiveness of implementing LAR in low-resource settings.

Minor Comments:

1. Add definitions for all abbreviations at first mention.

2. Include 95% CI for all metrics in Table 3 for consistency.

3. Improve labeling and color contrast in SHAP plots for readers unfamiliar with the method.

4. Address minor grammatical issues during copy-editing.

**Do you want your identity to be public for this peer review?** For information about this choice, including consent withdrawal, please see our Privacy Policy

Reviewer #1: No

Reviewer #2: **Yes: ** Kartini Hasballah

Reviewer #3: No

Reviewer #4: No

---

## [Author Response · Author response to Decision Letter 1]

28 Aug 2025

Response letter to reviewers

Dear Editors-in-Chief of the PLOS ONE,

We are very thankful for your insightful comments for our submitted manuscript.

MANUSCRIPT # PONE-D-25-41174, titled "Comparison of serum lactate and lactate-derived ratios as prognostic biomarkers in pediatric dengue shock syndrome using supervised machine learning models"

We are thankful for all peer-review points. These are our responses to all the reviewers’ comments.

Academic Editors:

Journal Requirements:

https://journals.plos.org/plosone/s/file?id=wjVg/PLOSOne_formatting_sample_main_body.pdf ;

Response: Yes, we are thankful for this comment. We have checked the journal format and have made appropriate amendments in the revised manuscript.

2. Please note that PLOS One has specific guidelines on code sharing for submissions in which author-generated code underpins the findings in the manuscript. In these cases, we expect all author-generated code to be made available without restrictions upon publication of the work.

Response: Thank you for this comment. We shared the code as appropriate request. The coding file has been submitted a as supplementary data (S2 Appendix).

3. We note that there is identifying data in the Supporting Information file <S1 Appendix.xlsx>. Due to the inclusion of these potentially identifying data, we have removed this file from your file inventory. Prior to sharing human research participant data, authors should consult with an ethics committee to ensure data are shared in accordance with participant consent and all applicable local laws.

Response: Thank you for this comment. As per your suggestion, we have removed the participants’ identifier (ID) from the submitted dataset in compliance with the anonymized principle. Revised S1 Appendix (dataset) has been amended and resubmitted, as appropriate.

Response: Thank you for this comment. We have followed your instructions.

Reviewer #1:

1. Title & Abstract — Please temper superlatives such as “excellent performance” unless supported by external validation and demonstrated clinical utility; present all performance metrics with 95% confidence intervals using consistent precision (for example, AUC 0.82, 95% CI 0.78–0.86) and include PR-AUC given class imbalance; explicitly list the composite outcome components and total event count in the abstract; and state the rule used to choose the “optimal” LAR cutoff (e.g., Youden index), reporting sensitivity and specificity with 95% CIs at that threshold.

Response: Thank you for your comment and pointing this out. We have rechecked this statement in the Abstract and we will make an appropriate language style based on your suggestion.

The optimal LAR cutoff was chosen based on the Youden index criteria.

We will report sensitivity and specificity with 95% CIs at that threshold in the revised Table below in the revised manuscript.

Table (Revised). Area under the receiver operating characteristic curve for serum lactate, lactate-to-albumin and lactate-to-bicarbonate ratios, and cut-off points for composite clinical endpoint among children with dengue shock syndrome

Parameters AUC of composite clinical endpoint (95% CI) p-value

Lactate (mmol/L) 0.72 (0.65 - 0.78) < 0.001

LAR 0.82 (0.76 - 0.87) < 0.001

L/B ratio 0.68 (0.62 - 0.74) < 0.01

Cutoff points Sensitivity Specificity LR (+) LR (-)

LAR ≥ 1.25 0.71

(0.60 – 0.80) 0.82

(0.78 – 0.86) 3.95

(3.35 – 4.65) 0.36

(0.29 – 0.44)

L/B ratio ≥ 0.20 0.56

(0.45 – 0.67) 0.75

(0.71 – 0.79) 2.26

(1.84 – 2.79) 0.58

(0.47 – 0.72)

AUC, Area under the curve; LR, Likelihood ratio, 95% CI, Confidence interval (AUC, sensitivity, specificity, LR+, LR-); LAR, Lactate-to-albumin ratio; L/B, Lactate-to-bicarbonate ratio

2. Introduction — Clarify the biological and clinical rationale for why LAR may outperform lactate or L/B in DSS, briefly contrasting prior 24-hour findings with admission-time measurements; define each component of the composite endpoint (death, mechanical ventilation, PALF, encephalitis) and justify their aggregation from a clinical-decision perspective; and close with a precise objective that commits to assessing both discrimination and calibration while comparing LAR against lactate and L/B.

Response: Thank you for this comment.

- Your suggestion of “Clarifying the biological and clinical rationale for why LAR may outperform lactate or L/B in DSS, briefly contrasting prior 24-hour findings with admission-time measurements” should be placed in the discussion section rather than introduction. We will answer your queries from point to point in the following.

2.1 LAR may outperform lactate or L/B in DSS:

In fact, we discussed the suggested point in the paragraph 5-in Discussion of submitted manuscript.

“Notably, the superior prognostic performance of the lactate-to-albumin ratio (LAR) underscores its relevance in capturing the complex and interrelated pathophysiological processes underlying severe dengue shock syndrome. As the denominator of the ratio, serum albumin serves as a surrogate marker for the severity of plasma leakage, which is the hallmark of dengue pathogenesis. Given its biological half-life of approximately three weeks [28], a rapid and profound decline in albumin levels in the acute phase of DSS cannot be attributed solely to impaired hepatic synthesis or increased catabolism. Rather, this decline reflects substantial redistribution of albumin secondary to dengue-induced endothelial injury. This process may be further compounded by hemodilution resulting from aggressive fluid resuscitation, which has been associated with adverse clinical outcomes [4, 29]. Furthermore, serum lactate, the numerator of LAR, indicates the metabolic consequences of plasma leakage, including impaired perfusion, anaerobic metabolism, and systemic circulatory failure. In DSS, elevated lactate levels not only signal the presence of shock but also reflect inadequate tissue oxygenation despite fluid resuscitation and vasopressor support. Therefore, LAR captures both the underlying pathophysiological insult (plasma leakage resulting in hypoalbuminemia) and its clinical consequences (refractory shock, evidenced by hyperlactatemia). This integrated metric provides a more comprehensive representation of dengue severity than either lactate or albumin alone. High LAR values may indicate excessive capillary leakage or evolving multi-organ failure, particularly in patients with dengue-associated acute liver failure who present late in the course of irreversible shock. Accordingly, the capacity of LAR to quantify the interplay between vascular pathology and systemic perfusion failure likely underlies its superior prognostic value in predicting critical outcomes in pediatric DSS.”

2.2 Briefly contrasting prior 24-hour findings with admission-time point:

Considering the pathogenesis of dengue shock syndrome, the acute progression of dengue severity was calculated by hours after dengue shock occurrence after PICU admission. Hence, at 24h after admission, lactate, albumin, and bicarbonate levels significantly changed, corresponding to the severity of dengue shock. Significantly, it should be acknowledged that serum LAR, LB, and lactate values may evolve over the first 24 h owing to fluid resuscitation, hemodilution, and ongoing vascular leakage, which may modify their predictive utility over time after hospitalization. Thus, the 24h values of such biomarkers became clearly overt compared to the baseline values. Therefore, at 24h biomarkers of LAR, LB ratio, and lactate had higher predictive values. Notably, our research team has recently published and shown a higher predictive value of serum lactate and LB ratio at 24h post-admission than at the time of admission in this study. This further suggests dynamic modelling of these biomarkers during the first 24h of PICU admission.

2.3 Define each component of the composite endpoint (death, mechanical ventilation, PALF, encephalitis) and justify their aggregation from a clinical-decision perspective; and close with a precise objective that commits to assessing both discrimination and calibration while comparing LAR against lactate and L/B.

We have clearly defined the component of the composite endpoint (death, mechanical ventilation, PALF, and encephalitis) in the submitted manuscript.

The rationale for using the composite endpoint is the low proportion of deaths (mortality of dengue cases). We have shown in our previous publications that mechanical ventilation requirement and dengue-associated acute liver failure are the two most significant causes contributing to high mortality rates in patients with severe dengue. Therefore, it is reasonable to combine MV and PALF into a composite endpoint (with death outcome).

The discrimination and calibration of the LAR- and LB-based predictive models have been additionally performed and are presented in the revised manuscript and in the following.

3. Methods: Study Design & Cohort — Add a transparent flow diagram detailing numbers screened, excluded with reasons, and included; specify inclusion criteria, timing of blood sampling relative to admission, handling of any repeated measures, and follow-up window; provide exact outcome definitions (including whether mechanical ventilation includes non-invasive ventilation) and per-component counts; and state whether a protocol existed or confirm that no prior registration was performed.

Response: Thanks for your insightful comments.

3.1 Add a transparent flow diagram detailing numbers screened, excluded with reasons, and included; specify inclusion criteria

Response: Your suggested points are important; however, it is a gentle reminder that this was a secondary dataset research of a published retrospective study. Obtaining your suggested data (timing of blood sampling relative to admission, handling of any repeated measures, and follow-up window) is challenging and impractical.

Therefore, we simplified the study flow diagram, according with the availability of secondary dataset, as shown in Fig 1. in the submitted manuscript.

Fig 1

Notably, we emphasized that this was a secondary data analysis, and we found that the main reason of exclusion is missing data of biomarkers (LAR, LB, Lactate). Thus, we determined to keep this study flow diagram as it is (Fig 1).

# Whether mechanical ventilation includes non-invasive ventilation:

Response: In this study, mechanical ventilation (MV) is exactly invasive MV.

Notably, nasal oxygenation and nasal continuous positive airway pressure (NCPAP) frequently used for DSS patients, were categorized as non-invasive ventilation (non-MV).

# State whether a protocol existed or confirm that no prior registration was performed.

Response: In this study, the national dengue diagnosis and management protocol (the Vietnamese Ministry of Health) has been applied consistently over the study period and so far.

4. Methods: Predictors & Missing Data — Provide a complete list of candidate predictors with units and admission-time availability and include a data dictionary as supplementary material; characterize missingness at the variable and patient levels and discuss the likely mechanism (MAR/MNAR), supported by a missingness heatmap; perform imputation within the resampling loop so that models only “see” imputed values created from their training folds; justify the >20% missingness exclusion rule, list excluded variables, and present a complete-case sensitivity analysis.

Response: We are thankful for your insightful comment. For missing data (count and percentage), S1 Fig. (Supplementary File) was submitted to the journal. We have answered your point by point and performed the missing data analysis below.

We included all patients with biomarkers (albumin, lactate, bicarbonate) in the analysis. Patients who lacked biomarker data were excluded.

Complete-case data analysis was applied for the sensitivity/specificity analysis of biomarkers. A complete-case sensitivity has been analyzed and submitted in the revised manuscript (Table 2).

For supervised machine-learning models, we used multiple-imputation data, as appropriate.

Mechanism of missing data in this study: Missing Completely at Random (MCAR)

To the best of our knowledge, there is no “black-and-white” principle to choose the 20%, 30%... missingness threshold for the exclusion rule. Hence, we determined >20% missingness as the threshold, because we mainly wanted the robust data for machine learning analysis to ensure the power and validity of study findings.

Based on your suggestion, we have added a missingness heatmap and Pareto chart (shown below), which will be submitted in the revision as supplementary data.

All predefined variables analyzed in the supervised models were complete (i.e no significant predictor in the supervised models was excluded).

5. Methods: Preprocessing & Modeling — Ensure that all preprocessing (scaling, one-hot encoding, class weighting or resampling, and any feature selection) is fit exclusively on training folds and then applied to held-out data; replace the single split with nested cross-validation or bootstrap optimism correction and, if feasible, add a temporal split (earlier years to train, later years to test) to probe transportability; report AUC, PR-AUC, Brier score, and expected calibration error with 95% CIs; include calibration-in-the-large and slope, add calibration plots, and consider isotonic or Platt recalibration; present decision-curve analysis to compare net benefit against treat-all/none policies and simple heuristics such as lactate alone; and show threshold-level confusion matrices, sensitivity, specificity, PPV, NPV, and likelihood ratios at clinically relevant cut points, specifying statistical tests (e.g., DeLong) and any multiplicity adjustments.

Response: Thank you for this comment. In current submission manuscript, we strictly performed most of what the reviewer mentioned above in data preprocessing and modeling. We ensure that all preprocessing (scaling, one-hot encoding, class weighting or resampling, and any feature selection) is fit exclusively on training folds and then applied to held-out data; the single split with nested cross-validation or bootstrap optimism correction. For the confusion matrices, statistical tests were performed using the DeLong method. Additionally, no interactions between the clinical predictors and biomarkers were found.

We performed additional analyses of the calibration plot and decision curve, and submitted these data in the revised manuscript. These data are shown below in question 6-Result section.

6. Results: Cohort & Outcomes — Provide a baseline characteristics table stratified by outcome with standardized mean differences to convey distributional separation; report per-component outcome counts and, where possible, univariable associations of LAR, lactate, and L/B with each component; when proposing cutoffs, give sensitivity/specificity with CIs and LR+/LR− alongside prevalence-adjusted PPV/NPV to guide bedside interpretation; and for machine-learning models, present outer-fold performance (mean ± SD), calibration results, and decision-curve plots.

Response: As the reviewer suggested, we further perform analysis of standardized mean differences using R package

---

## [Decision Letter · Decision Letter 1]

13 Sep 2025

Dear Dr. Thanh ,

Thank you for submitting your manuscript to PLOS ONE. After careful consideration, we feel that it has merit but does not fully meet PLOS ONE’s publication criteria as it currently stands. Therefore, we invite you to submit a revised version of the manuscript that addresses the points raised during the review process.

We look forward to receiving your revised manuscript.

Kind regards,

Muhammad Iqhrammullah, Ph.D

Academic Editor

PLOS ONE

Journal Requirements:

Reviewers' comments:

Reviewer's Responses to Questions

**Comments to the Author**

Reviewer #1: (No Response)

Reviewer #4: All comments have been addressed

2. Is the manuscript technically sound, and do the data support the conclusions?

Reviewer #1: Yes

Reviewer #4: Yes

3. Has the statistical analysis been performed appropriately and rigorously?

Reviewer #1: Yes

Reviewer #4: Yes

4. Have the authors made all data underlying the findings in their manuscript fully available?

Reviewer #1: (No Response)

Reviewer #4: Yes

5. Is the manuscript presented in an intelligible fashion and written in standard English?

Reviewer #1: Yes

Reviewer #4: Yes

Reviewer #1: (No Response)

Reviewer #4: The authors have addressed all previous reviewer comments comprehensively. The abstract has been revised for clarity, the statistical analyses have been expanded with calibration and decision-curve analyses, and the presentation of baseline characteristics has been improved. The manuscript is technically sound, with appropriate methodology, robust validation, and clear reporting of results.

The inclusion of SHAP-based model interpretability and additional analyses of missingness strengthen the reliability of the findings. The conclusions are well supported by the data and provide meaningful clinical implications for the use of lactate-derived ratios in pediatric dengue shock syndrome.

The English language is clear and intelligible, with only minor stylistic adjustments potentially beneficial but not essential. Overall, the manuscript has been substantially improved and is now suitable for publication in PLOS ONE.

**Do you want your identity to be public for this peer review?** For information about this choice, including consent withdrawal, please see our Privacy Policy

Reviewer #1: No

Reviewer #4: No

---

## [Author Response · Author response to Decision Letter 2]

14 Sep 2025

Response letter to the journal editors

Dear Editors-in-Chief of the PLOS ONE,

We are very thankful for your comments on our submitted manuscript.

MANUSCRIPT # PONE-D-25-41174R1

“Comparison of serum lactate and lactate-derived ratios as prognostic biomarkers in pediatric dengue shock syndrome using supervised machine learning models”

These are our responses to all the reviewers’ comments.

Journal Requirements:

Response: Thank you for this comment. We have made appropriate amendments in the revised manuscript.

We have meticulously looked through the previously reference list and determined to remove these cited references, in terms of less relevance, as follows:

World Health Organization‎. Global strategy for dengue prevention and control 2012-2020. World Health Organization, 2012. [Accessed and cited on 04th December 2023]. Available from: https://iris.who.int/handle/10665/75303

Nguyen TT, Nguyen DT, Vo TT, Dang OT, Nguyen BT, Pham DT, et al. Associations of obesity and dengue-associated mortality, acute liver failure and mechanical ventilation in children with dengue shock syndrome. Medicine (Baltimore). 2023; 102(46):e36054. doi: 10.1097/MD.0000000000036054. PMID: 37986332; PMCID: PMC10659721

Nguyen TT, Le NT, Nguyen NM, Do VC, Trinh TH, Vo LT. Clinical features and management of children with dengue-associated obstructive shock syndrome: A case report. Medicine (Baltimore). 2022; 101(43):e31322. doi: 10.1097/MD.0000000000031322. PMID: 36316901

Vo LT, Vu T, Pham TN, Trinh TH, Nguyen TT. Machine learning-based models for prediction of in-hospital mortality in patients with dengue shock syndrome. World J Methodol. 2025; 15(3): 101837. [DOI: 10.5662/wjm.v15.i3.101837]

Response: Thank you for your comment. We have removed retracted articles from reference and replaced appropriate references as suggested.

We have carefully checked the retraction status of updated cited references in the revision, and we affirm that all cited references were checked for retraction or expressions of concern (via PubMed/PMC, publisher webpages and CrossRef on 14 September 2025; no retractions or expressions of concern have been identified under current submission).

3. “While revising your submission, please upload your figure files to the Preflight Analysis and Conversion Engine (PACE) digital diagnostic tool, https://pacev2.apexcovantage.com/. PACE helps ensure that figures meet PLOS requirements.”

Response:

Because we are not familiar with using PACE, considering the journal’s suggestion, we kindly suggest the journal editors to do a favor to help our research team with this task, editing all figures to meer the journal requirement for publications. All the figures are now in 600 dpi format.

We request assistance in formatting the figures (width and length) to meet the journal’s standards.

We have made the appropriate amendments as per your suggestions.

We look forward to your feedback and answer any further questions.

Many thanks,

Best regards,

Dr. Nguyen Tat Thanh

---

## [Decision Letter · Decision Letter 2]

24 Sep 2025

Dear Dr. Thanh ,

Thank you for submitting your manuscript to PLOS ONE. After careful consideration, we feel that it has merit but does not fully meet PLOS ONE’s publication criteria as it currently stands. Therefore, we invite you to submit a revised version of the manuscript that addresses the points raised during the review process.

Please carefully consider all comments from the reviewers. I agree with Reviewer 4 that the limitation paragraph can still benefit of further elaboration, especially considering the data collection method. In addition, kindly revise the "data availability" statement and cite the primary data source: https://journals.plos.org/plosone/article?id=10.1371/journal.pone.0126134

We look forward to receiving your revised manuscript.

Kind regards,

Muhammad Iqhrammullah, Ph.D

Academic Editor

PLOS ONE

Journal Requirements:

Reviewers' comments:

Reviewer's Responses to Questions

**Comments to the Author**

Reviewer #1: All comments have been addressed

Reviewer #4: All comments have been addressed

2. Is the manuscript technically sound, and do the data support the conclusions?

Reviewer #1: Yes

Reviewer #4: Yes

3. Has the statistical analysis been performed appropriately and rigorously?

Reviewer #1: Yes

Reviewer #4: Yes

4. Have the authors made all data underlying the findings in their manuscript fully available?

Reviewer #1: Yes

Reviewer #4: Yes

5. Is the manuscript presented in an intelligible fashion and written in standard English?

Reviewer #1: Yes

Reviewer #4: Yes

Reviewer #1: After the second revision, all concerns and comments have been adequately addressed by the authors, therefore I state no further comments. Props to the author.

Reviewer #4: Strengths:

1. Important clinical relevance: pediatric DSS with high mortality risk.

2. Large cohort (524 patients) with comprehensive clinical and laboratory data.

3. Novelty: evaluation of lactate-derived ratios (LAR, LB) using machine learning, which is underexplored in pediatrics.

4. Robust methodology with multiple supervised models, SHAP interpretability, and decision curve analysis.

5. Clear finding that LAR outperforms lactate and LB ratio for prognostic prediction.

Points for minor revision:

1. Clarity of language: Some sentences in the Introduction and Discussion are long and complex. Please consider a professional language edit to improve readability for an international audience.

2. Logistic regression overfitting: In the Results, you note potential overfitting of LR. It would be useful to briefly explain how this may affect interpretation and why ensemble models are more reliable here.

3. Figures: You requested editorial assistance for figure formatting. If possible, ensure figure legends are fully self-explanatory, particularly for Fig 2 (RF variable importance) and Fig 3 (SHAP summary plot).

4. Limitations: The Discussion could be slightly expanded to emphasize that this is a single-center retrospective analysis, and that external validation in independent pediatric cohorts is necessary.

5. Clinical applicability: While the statistical performance of LAR is strong, please comment on how this could be practically implemented in resource-limited DSS treatment settings.

Overall, this is a strong and clinically impactful paper. With minor polishing, it will make a valuable contribution.

**Do you want your identity to be public for this peer review?** For information about this choice, including consent withdrawal, please see our Privacy Policy

Reviewer #1: No

Reviewer #4: No

---

## [Author Response · Author response to Decision Letter 3]

27 Sep 2025

Response letter to reviewers

Dear Editors-in-Chief of the PLOS ONE,

We are very thankful for your insightful comments for our submitted manuscript.

MANUSCRIPT # PONE-D-25-41174, titled "Comparison of serum lactate and lactate-derived ratios as prognostic biomarkers in pediatric dengue shock syndrome using supervised machine learning models"

We are thankful for all peer-review points. These are our responses to all the reviewers’ comments.

Journal Editors:

Please carefully consider all comments from the reviewers. I agree with Reviewer 4 that the limitation paragraph can still benefit of further elaboration, especially considering the data collection method.

In addition, kindly revise the "data availability" statement and cite the primary data source: https://journals.plos.org/plosone/article?id=10.1371/journal.pone.0126134

Response: We are thankful for your comments. We have made appropriate amendments according to your suggestions.

Data availability: All relevant data are within the paper and its Supporting Information files.

Our primary data source:

All relevant data supporting the findings of this study are available within the paper, its Supporting Information files, and the primary dataset from the parent study: Nguyen Tat T, Vo Hoang-Thien N, Nguyen Tat D, Nguyen PH, Ho LT, Doan DH, et al. Prognostic values of serum lactate-to-bicarbonate ratio and lactate for predicting 28-day in-hospital mortality in children with dengue shock syndrome. Medicine (Baltimore). 2024;103(17):e38000. doi: 10.1097/MD.0000000000038000. PMID: 38669370; PMCID: PMC11049702. This dataset was approved by the Scientific Committee and Institutional Review Board of Children’s Hospital No.2, Ho Chi Minh City, Vietnam (IRB No. 893/QD-BVND2; 06 June 2022).

We have cited this reference as appropriate.

[4] Nguyen Tat T, Vo Hoang-Thien N, Nguyen Tat D, Nguyen PH, Ho LT, Doan DH, et al. Prognostic values of serum lactate-to-bicarbonate ratio and lactate for predicting 28-day in-hospital mortality in children with dengue shock syndrome. Medicine (Baltimore). 2024 ;103(17):e38000. doi: 10.1097/MD.0000000000038000. PMID: 38669370; PMCID: PMC11049702.

The limitation paragraph can still benefit of further elaboration, especially considering the data collection method.

Response: We thank for your comment. We have made appropriate amendments according to your suggestions in the revised manuscript (Lines 584-588).

“As this study was based on secondary data analysis, it was constrained by the scope and quality of the information collected in the original investigation, with no possibility of verifying or supplementing missing variables. Furthermore, the consistency of the recorded data may have been affected by variations in clinical practice and documentation during the study period.”

Journal Requirements:

Response: Thank you for your comment. We have strictly followed this recommendation.

Response: Thank for your comment. We have strictly followed this recommendation. We have checked and no retracted articles have been found in the reference list.

Reviewer #4:

Strengths:

1. Important clinical relevance: pediatric DSS with high mortality risk.

2. Large cohort (524 patients) with comprehensive clinical and laboratory data.

3. Novelty: evaluation of lactate-derived ratios (LAR, LB) using machine learning, which is underexplored in pediatrics.

4. Robust methodology with multiple supervised models, SHAP interpretability, and decision curve analysis.

5. Clear finding that LAR outperforms lactate and LB ratio for prognostic prediction.

Points for minor revision:

1. Clarity of language: Some sentences in the Introduction and Discussion are long and complex. Please consider a professional language edit to improve readability for an international audience.

2. Logistic regression overfitting: In the Results, you note potential overfitting of LR. It would be useful to briefly explain how this may affect interpretation and why ensemble models are more reliable here.

3. Figures: You requested editorial assistance for figure formatting. If possible, ensure figure legends are fully self-explanatory, particularly for Fig 2 (RF variable importance) and Fig 3 (SHAP summary plot).

4. Limitations: The Discussion could be slightly expanded to emphasize that this is a single-center retrospective analysis, and that external validation in independent pediatric cohorts is necessary.

5. Clinical applicability: While the statistical performance of LAR is strong, please comment on how this could be practically implemented in resource-limited DSS treatment settings.

Overall, this is a strong and clinically impactful paper. With minor polishing, it will make a valuable contribution.

Response:

We are thankful for the reviewers’ comments for making our paper more comprehensive to the readers. These are our responses to your comments.

1. Clarity of language: Some sentences in the Introduction and Discussion are long and complex. Please consider a professional language edit to improve readability for an international audience.

Response: We are thankful for your comments, and we have made appropriate amendments in the revised manuscript.

2. Logistic regression overfitting: In the Results, you note potential overfitting of LR. It would be useful to briefly explain how this may affect interpretation and why ensemble models are more reliable here.

Response: We are thankful for your comments. In our opinion, the Results section should mainly describe what the statistical analyses yield the results. In this section, it is not recommended to explain or discuss any thing about data results. Instead, explanation (or interpretation study results) should be placed in the Discussion section.

In this respect, we have discussed the logistic regression overfitting and showed that ensemble models are more reliable with appropriately cited references in the paragraph 4, Discussion section.

3. Figures: You requested editorial assistance for figure formatting. If possible, ensure figure legends are fully self-explanatory, particularly for Fig 2 (RF variable importance) and Fig 3 (SHAP summary plot).

Response: We are thankful for your comments. Appropriate figure legends have been provided in the revised manuscript as well. Notably, we aim to provide a succinct and clear description, and our target readers are mainly researchers with a good background in mathematical modelling.

# Previous submission version

[Fig 2 Caption]

Fig 2. Comparisons of serum lactate levels, lactate-derived ratios, and significant clinical variables of the random forest model based on classification accuracy.

Revision

Fig 2. Key predictors of the composite clinical endpoint in dengue shock syndrome identified by the random forest model.

The lactate-to-albumin ratio (LAR) was the strongest predictor, outperforming serum lactate and the lactate-to-bicarbonate ratio. Additional important variables included markers of hemodynamic compromise, organ dysfunction, and disease severity, reflecting the multifactorial pathophysiology of dengue shock syndrome.

# Previous submission version

[Fig 3 Caption]

Fig 3. SHAP summary plot illustrating the relative contributions of serum lactate, lactate-derived ratios (LAR and LB), and clinical features to the prediction of critical outcomes in pediatric patients with dengue shock syndrome.

Revision

Fig 3. SHAP summary plot of predictors of critical outcomes in pediatric dengue shock syndrome.

SHAP analysis identified elevated INR, platelet transfusion requirement, high fluid administration, and respiratory failure as the strongest clinical predictors, whereas LAR was the most informative biomarker, followed by LB ratio and serum lactate. These results emphasize the prognostic utility of serum lactate-derived indices and the importance of multiple clinical features in determining the outcomes.

4. Limitations: The Discussion could be slightly expanded to emphasize that this is a single-center retrospective analysis, and that external validation in independent pediatric cohorts is necessary.

Response: We did discuss these points and we have provided this information in our previous manuscript sunbmitted to the journal.

Lines 579-586, Paragraph 7, Discussion section

Lines 604-605, last sentences in the Paragraph 8 (last paragraph), Discussion section

We have clearly states these points in the previous submission. Importantly, the readers of this niche paper have a solid background in scientific research; hence, we believe there is no need to elaborate these information in so details. Thank you.

5. Clinical applicability: While the statistical performance of LAR is strong, please comment on how this could be practically implemented in resource-limited DSS treatment settings.

Response: We did discuss and provide this information in our previous manuscript sunbmitted to the journal in Paragraph 8 (last paragraph), Discussion section. Per your suggestion, we make a minor amendment for clarification in the revision (last paragraph, Discussion section).

Revision

This study highlights the potential of lactate-derived biomarkers, particularly the lactate-to-albumin ratio (LAR), as practical tools for early risk stratification in pediatric dengue shock syndrome. Compared with the lactate-to-bicarbonate (LB) ratio and serum lactate alone, LAR demonstrated superior prognostic performance in predicting critical complications, including mechanical ventilation, acute liver failure, and dengue-associated encephalitis. Importantly, both serum lactate and albumin tests are routinely available and low-cost laboratory measurements in most provincial and district hospitals, making the LAR feasible for implementation, even in resource-limited settings. Incorporating LAR into admission triage protocols may allow for the earlier identification of high-risk children, support timely referral to higher-level facilities, and guide the prioritization of intensive monitoring and therapeutic interventions. Nevertheless, prospective validation across diverse healthcare systems is needed to confirm its generalizability before it can be adopted as a standard risk assessment tool.

We have answered all the questions by all the reviewers and have offered corrections as needed. We are thankful for your significant comments that have made our study more comprehensive to the readers.

We look forward to your feedback and answer any further questions.

Many thanks,

Best regards,

Dr Thanh Nguyen Tat

---

## [Editor Report · Decision Letter 3]

6 Oct 2025

Comparison of serum lactate and lactate-derived ratios as prognostic biomarkers in pediatric dengue shock syndrome using supervised machine learning models

PONE-D-25-41174R3

Dear Dr. Thanh ,

We’re pleased to inform you that your manuscript has been judged scientifically suitable for publication and will be formally accepted for publication once it meets all outstanding technical requirements.

Kind regards,

Muhammad Iqhrammullah, Ph.D

Academic Editor

PLOS ONE
---

## [Editor Report · Acceptance letter]

PONE-D-25-41174R3

PLOS ONE

Dear Dr. Thanh ,

I'm pleased to inform you that your manuscript has been deemed suitable for publication in PLOS ONE. Congratulations! Your manuscript is now being handed over to our production team.

Kind regards,

on behalf of

Dr. Muhammad Iqhrammullah

Academic Editor

PLOS ONE